# Comparing retinotopic maps of children and adults reveals a late-stage change in how V1 samples the visual field

**Marc M. Himmelberg** [1,2] ✉, **Ekin Tünçok** [1], **Jesse Gomez** [3],
**Kalanit Grill-Spector** [4,5], **Marisa Carrasco** [1,2,6] & **Jonathan Winawer** [1,2,6]

Adult visual performance differs with angular location –it is better for stimuli along the horizontal than vertical, and lower than upper vertical meridian of the visual field. These perceptual asymmetries are paralleled by asymmetries in cortical surface area in primary visual cortex (V1). Children, unlike adults, have similar visual performance at the lower and upper vertical meridian. Do children have similar V1 surface area representing the upper and lower vertical meridian? Using MRI, we measure the surface area of retinotopic maps (V1-V3) in children and adults. Many features of the maps are similar between groups, including greater V1 surface area for the horizontal than vertical meridian. However, unlike adults, children have a similar amount of V1 surface area representing the lower and upper vertical meridian. These data reveal a late-stage change in V1 organization that may relate to the emergence of the visual performance asymmetry along the vertical meridian by adulthood.

A central question in systems neuroscience is whether and how the organization of cortical maps changes across the lifespan. Some properties of human early visual cortex mature early; functional MRI studies show that visual field map surface area[1], population receptive field (pRF) sizes[1,2], and the change in cortical magnification with eccentricity[1,3] are adult-like by childhood (6–12 years). In contrast, performance on some basic visual tasks such as vernier acuity[4], contour integration[5,6], and texture segmentation[7] do not mature until adolescence (13 + years)[8] and are likely to be limited by the circuitry of primary visual cortex (V1)[9–11]. Whereas many properties of primary sensory cortices, including V1, mature early in development[12–14], these behavioral findings suggest that V1 likely develops across a longer period than shown by fMRI studies.

One striking difference in visual performance between adults and children pertains to polar angle asymmetries. Both adults and children show better visual performance for stimuli along the horizontal than vertical meridian (horizontal-vertical asymmetry; HVA) at a matched eccentricity[15]. However, only adults show better visual performance for

stimuli along the lower than upper vertical meridian (vertical meridian anisotropy; VMA) for a variety of perceptual tasks (e.g., orientation discrimination, contrast sensitivity, and acuity[16–22]). Instead, children do not have a VMA; their visual performance is similar between the upper and lower vertical meridian[15]. In adults, the perceptual polar angle asymmetries are well matched to the distribution of cortical tissue in V1; there is more V1 tissue representing the horizontal than vertical meridian, and the lower than upper vertical meridian of the visual field[23–26]. For review see ref. [27]. These cortical polar angle asymmetries have not been quantified in children. Because the visual performance asymmetry along the vertical meridian differs between adults and children, we hypothesize that the cortical asymmetry in the amount of V1 tissue representing the vertical meridian will also differ between adults and children.

Here, we quantified and compared the distribution of cortical surface area representing the visual field between adults and children. We used fMRI-based population receptive field (pRF) modeling to measure retinotopic maps in early visual cortex (V1, V2, and V3) in 24

[1]Department of Psychology, New York University, New York, NY 10003, USA. [2]Center for Neural Science, New York University, New York, NY 10003, USA. [3]Princeton Neuroscience Institute, Princeton University, Princeton, NJ 08540, USA. [4]Department of Psychology, Stanford University, Stanford, CA 94305, USA. [5]Wu Tsai Neurosciences Institute, Stanford University, Stanford, CA 94305, USA. [6]These authors jointly supervised this work: Marisa Carrasco, Jonathan Winawer. ✉e-mail: marc.himmelberg@nyu.edu

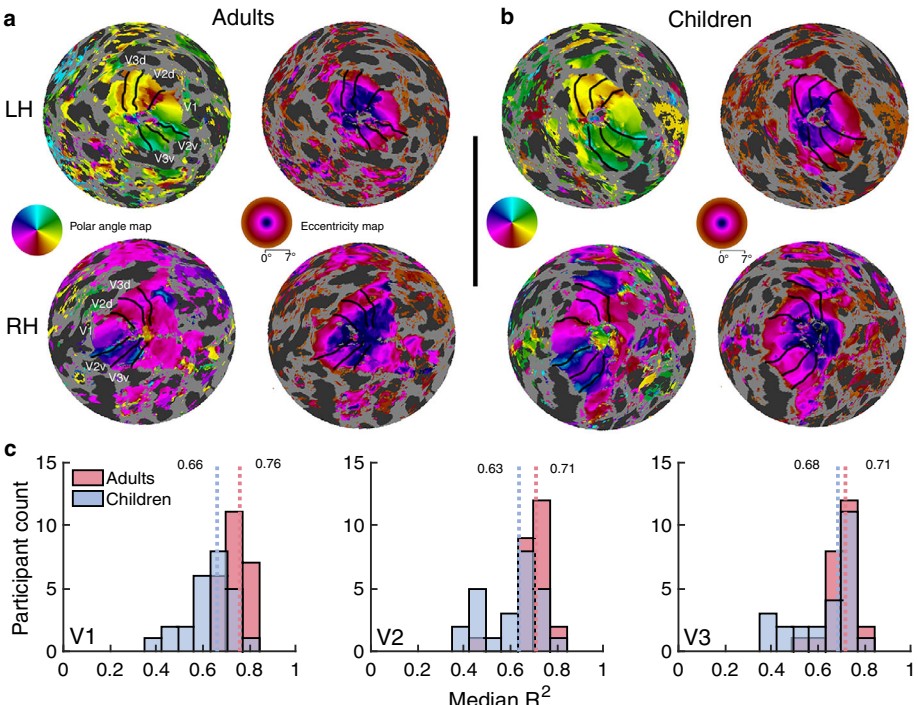

**Fig. 1 | Example retinotopic maps and the median variance explained for adults and children.** Examples of polar angle and eccentricity maps (0–7°) from (**a**) two adults and (**b**) two children. Each hemisphere comes from a unique participant (LH = Left hemisphere, RH = Right hemisphere). Retinotopic maps are projected onto cortical flatmaps of the *fsnative* surface with the occipital pole in the center of the map. V1, V2, and V3 boundaries are indicated as black lines. Inset legends show polar angle and eccentricity maps. Data are thresholded at $R^2 \geq 0.10$. **c** Histograms of the median variance explained ($R^2$) of V1, V2, and V3 for adults (red) and children (blue). Dashed red and blue lines and the values beside them show the group-level median $R^2$ for adults and children, respectively. Source data are provided as a Source Data file.

adults (22-26 years old) and 25 children (5-12 years old). We compared the following metrics of the cortical representation of the visual field: 1) the surface area of the V1, V2, and V3 maps; 2) areal cortical magnification (mm² surface area per 1° of visual angle) as a function of eccentricity within each of V1, V2, and V3; and 3) cortical surface area as a function of polar angle in V1, comparing the representation of the horizontal and vertical meridians, and the lower and upper vertical meridians of the visual field. Based on prior work, we hypothesized that the first two measures would be similar between adults and children, whereas the third would differ (i.e., adults would have a cortical VMA, whereas children would not) given the different pattern of visual performance along the vertical meridian between adults and children[15].

Our data show that within V1, V2, and V3, map surface area and cortical magnification as a function of eccentricity are similar between adults and children. We also find that both adults and children have greater V1 surface area representing the horizontal than vertical meridian of the visual field, consistent with better visual performance along the horizontal than vertical meridian in both groups[15–21]. However, whereas adults have more V1 surface area representing the lower than the upper vertical meridian (in agreement with prior studies[23–26]), children have no difference, consistent with their visual performance at these locations. This difference in the cortical representation of the vertical meridian between adults and children reveals a surprisingly large late-stage change in how V1 samples the visual field and parallels the emergence of a vertical meridian asymmetry in visual performance by adulthood.

## Results

### High quality retinotopic maps measured in adults and children
First, a series of quality checks were run on the retinotopy data to determine the quality of the maps in adults and children. Our primary research questions concern the surface area of either whole visual maps or parts of these maps. The boundaries of these maps are defined by polar angle features, rather than by variance explained or signal-to-noise (SNR). Thus, the surface area estimates do not systematically differ with data quality –any systematic differences in data quality between adults and children would not influence surface area measurements, as long as the data are of sufficient quality to delineate the maps in each participant. Hence, we assessed the overall quality of the retinotopic maps for each group, rather than comparing map quality between adults and children.

In adults and children, visual inspection of the retinotopic maps indicated clear polar angle and eccentricity representations that were organized as expected for V1, V2, and V3. Examples of polar angle and eccentricity maps for two adults and two children are illustrated in Fig. 1a, b and additional maps on inflated surfaces are available in prior work using the same dataset[2] and in subsequent figures in this paper. To complement the subjective assessment of the maps, for each participant, we computed the median variance explained ($R^2$) of the BOLD time series by the pRF fits for vertices within V1, V2, and V3 (Fig. 1c). The median $R^2$ of the pRF fits (indicated by the dashed vertical lines in Fig. 1c) were high ($R^2 > 0.60$) in all maps for both adults and children, indicating quality data. Both the large-scale structure evident in the retinotopic maps and the high variance explained for adults and children make it clear that the data are sufficient to delineate visual field boundaries and make measurements of cortical surface area.

Prior work has shown that the left and right hemispheres of retinotopic maps are similar in size[28,29]. Thus, an additional metric of data quality is the correlation of the surface area of the left and right hemispheres of the maps. This metric depends on the entire processing pipeline: from data acquisition, to pRF model fitting, to manual delineation of the ROIs, as noise at any stage of the pipeline could

lower the correlation between hemispheres. A high correlation would indicate that the ROI definitions are of sufficient quality to be used for estimates of map surface area. For adults and children, the surface area of the left and right hemispheres of V1, V2, and V3 were all highly correlated (Fig. 2). The high level of symmetry indicates that the map definitions are of good quality for both adults and children.

### The surface area of early visual field maps is similar for adults and children

We quantified the overall surface area (mm²) of the V1, V2, and V3 maps within the 0–7° eccentricity range defined by our stimulus extent. The central 7° is 38% of the entire V1 map, according to the formula for areal cortical magnification proposed by Horton and Hoyt (1991)[30]. The surface areas of each map were summed across the left and right hemisphere within an individual participant.

First, we compared the mean surface area of the visual field maps for adults and children. The surface areas of V1, V2, and V3 maps were nearly the same in adults and children, whether measured as median (Fig. 3a) or mean (Table 1), differing only by a few percent. On average, bootstrapping of the group means (1000 bootstraps) showed that the maps were slightly larger in adults than children; V1 was 5.5% larger (CI$^{95}$ = [5.1%, 6.0%]), V2 was 3.7% larger (CI$^{95}$ = [3.2%, 3.9%]), and V3 was

4.3% larger (CI$^{95}$ = [4.0%, 4.6%]). For both adults and children, V1 was similar in size to V2, and V3 was smaller than V1 and V2. Further, because the total surface area of the cortex differed only slightly between adults and children (4.3% larger for adults, CI$^{95}$ = [4.2%, 4.5%]) (Fig. 3b), the means remained similar between groups after normalizing the surface area of each participant's visual map by the total surface area of their cortex.

We assessed sex differences in the surface area of the maps and found little difference in the surface area of V1, V2, or V3 when comparing between male and female adults, male and female children, or when the surface area of each individual's map was normalized to the total surface area of their cortex for adults and children (all $p > 0.100$, two-tailed independent t-tests).

We then compared the variability in map surface area between the two groups. We calculated the coefficient of variation ($\sigma / \mu$) for each of the three maps across participants (Table 1). Variability in the surface area of the maps was similar for adults and children. Notably, the surface area of adult V1 varied more than twofold, consistent with prior measurements[24,28,29]; the smallest V1 was 1367 mm² and the largest was 3206 mm². Similar variability in V1 surface area was found in children; the smallest V1 was 1257 mm² and the largest was 3312 mm². The coefficient of variation was also similar for the groups in V2 and V3. Notably, the variation within groups (a CV of ~20%) is much larger than the small differences in means between groups (~3–5%).

### Cortical magnification as a function of eccentricity in early visual field maps is similar between adults and children

The amount of cortical surface area dedicated to processing the visual field decreases with increasing eccentricity from the center of visual space[31,32] which can be quantified by the cortical magnification function[33]. We calculated areal cortical magnification (mm² of surface area/deg² of visual space) as a function of eccentricity –between 1° and 7°– for V1, V2, and V3. Areal cortical magnification was calculated by summing the surface area of vertices whose pRF centers fell within an eccentricity ring in visual space (see Methods: Cortical magnification as a function of eccentricity). The summed surface area was then divided by the area of visual space encapsulated by the eccentricity ring to calculate mm² of surface area per degree² of visual space. This eccentricity ring was stepped across the cortical surface, from 1° to 7° of eccentricity. The ring began at 1° eccentricity rather than 0° because noise in the pRF estimates of retinotopic coordinates near the foveal confluence tends to be large[23,29,34] and because the fixation task covered the central 0.5° of the display during the pRF mapping procedure. As the cortical magnification calculation is a fraction –with degree of visual space in the denominator– noisy estimates of small values will have a large effect on the computed fraction.

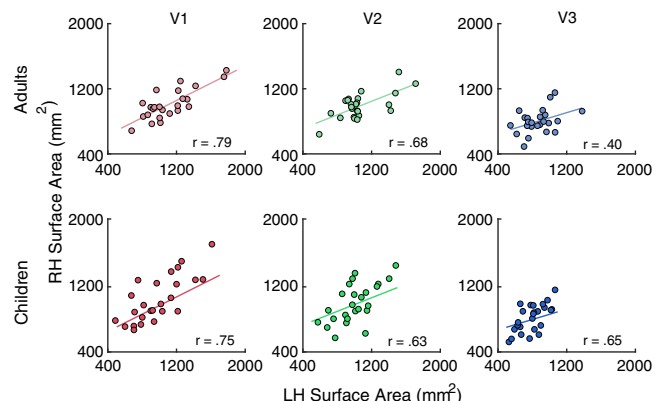

**Fig. 2 | Correlations of the surface area between left and right hemispheres of early visual field maps.** The colored line represents a line of best fit through the data showing the positive correlation between the left and right hemispheres of each hand-drawn ROI for adults ($n = 24$) and children ($n = 25$). Pearson's correlations (two-tailed) found correlations between the left and right hemisphere in adult V1 ($r = .79$, $p = 0.001$), V2 ($r = 0.68$, $p = 0.001$), and V3 ($r = 0.40$, $p = 0.025$). The same was found for child V1 ($r = 0.75$, $p = 0.001$), V2 ($r = 0.63$, $p = 0.001$), and V3 ($r = 0.65$, $p = 0.001$). Source data are provided as a Source Data file.

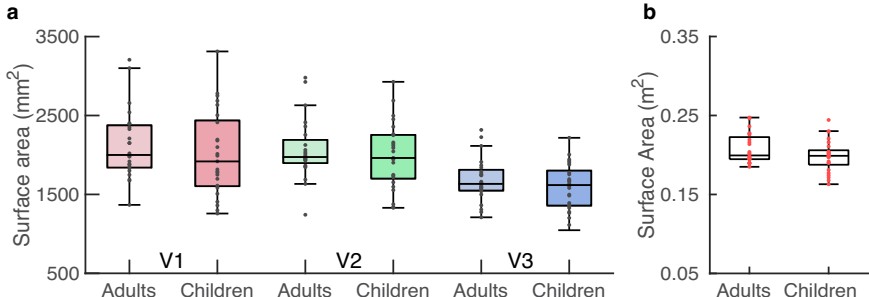

**Fig. 3 | Measurements of the surface area of early visual field maps and the cortical sheet. a** Surface area (millimeters²) of V1, V2, and V3 for adults ($n = 24$) and children ($n = 25$). Individual data are plotted as black data points and the colored horizontal line represents the group-median surface area. Because the surface areas are summed across hemispheres, the values are about double those in Fig. 2. **b** The total surface area (meters²) of the cortical sheet for adults ($n = 24$) and children

($n = 25$). Individual data are plotted as red data points and the black horizontal line represents the median surface area. For both (**a**) and (**b**), the top and bottom edges of each box represent the 75th and 25th percentiles, respectively, and the whiskers extend to the maxima and minima data points not considered outliers. The y-axes are matched so that the scaling in (**b**) is 100x the size of that in (**a**). Source data are provided as a Source Data file.

In V1, V2, and V3, cortical magnification decreased as a function of eccentricity for both groups, as expected (Fig. 4). In V1, for eccentricities above 1.5°, the cortical magnification function for both adults and children was close to the estimates derived from the adult lesion data from Horton and Hoyt (1991)[30]. The estimates below 1.5° are noisier for the reasons described above. Overall, the cortical magnification function was highly similar between adults and children in all three maps. There is one subtle difference between groups: in these maps, especially V2, adults had slightly more surface area representing central parts of the visual field than children. We return to this point in the Discussion.

### Polar angle asymmetries in V1 surface area differ between adults and children

Next, we examined polar angle asymmetries in V1 surface area. We focused on V1 rather than V2 and V3 because V1 is a large, continuous map of the contralateral hemifield, whereas V2 and V3 are split into quarterfields thereby making it difficult to robustly assess surface area as a function of fine changes in polar angle (though see Silva et al. (2018)[26] for measures from V2/V3 in adults). Cortical polar angle asymmetries have previously been identified in adult V1; there is more V1 surface area (thus cortical tissue) representing the horizontal than vertical meridian, and the lower than upper vertical meridian of the visual field[23–26]. Here, we quantified these cortical polar angle asymmetries in children. For each participant, we calculated the amount of V1 surface area representing angular wedge-ROIs. These contiguous wedge-ROIs were defined in visual space using pRF centers and were centered on the left and right horizontal, upper vertical, and lower vertical meridians of the visual field. The wedge-ROIs gradually increased in width, from ±15° to ±55°, and spanned 1–7° of eccentricity (see Methods: Cortical magnification as a function of polar angle for details on how pRF centers were used to generate these wedge-ROIs in visual space). This eccentricity range was chosen to exclude the central 1° which can include noisy polar angle position estimates.

First, we examined the adult data and confirmed a cortical horizontal-vertical anisotropy (HVA) and vertical-meridian asymmetry (VMA). Specifically, there was more V1 surface area representing the horizontal than vertical meridian (Fig. 5a), and the lower than upper vertical meridian of the visual field (Fig. 5b). These polar angle asymmetries were large and were consistent with increasing wedge-ROI width up to ±55°. These measurements were made on the midgray surface. Given that surface area calculations depend on the cortical depth used to derive the surface area of each vertex, we also conducted these analyses on the pial and white matter surfaces and similarly identified a cortical HVA and VMA (Supplementary figs. 1, 2).

We then quantified V1 surface area as a function of polar angle in children. We identified a cortical HVA in children; there was more V1 surface area representing the horizontal than vertical meridian (Fig. 5c). The effect was robust and similar to that observed in adults. However, children did not have a cortical VMA. Unlike adults, children had a similar amount of surface area representing the lower versus the upper vertical meridian (Fig. 5d). Repeating these analyses on the pial and white matter surfaces similarly identified a cortical HVA but no VMA in children (Supplementary Figs. 1 and 2). For completeness, we also plotted estimates of cortical thickness (Supplementary Fig. S3), surface curvature (Supplementary Fig. 4), and pRF size (Supplementary Fig. 5) as a function of polar angle.

In a complementary analysis, we measured the surface area of V1 dedicated to portions of the visual field without enforcing contiguous wedge-ROIs. This method is computationally much simpler than the wedge-ROI approach but is subject to noise. We did this by summing the surface area of V1 vertices with pRF centers between 1° and 7° of eccentricity within increasing polar angle ranges (after thresholding by 10% variance explained). These calculations do not rely on smoothing or templates, nor the assumption of continuity in the V1 map. Like the wedge-ROI method, this simpler method shows a large HVA in both groups and a large VMA in adults; children show a weak VMA, less than half of that measured in adults (Supplementary Fig. 6).

### Children have less surface area representing the lower vertical meridian than adults

These data show a distinct difference between adults' and children's cortical representation of the vertical meridian. Unlike adults, children have no cortical VMA; there is no difference in the amount of V1 surface area representing the lower and upper vertical meridian. The data in Fig. 5d show a pattern of children having less surface area representing the lower vertical meridian than adults, whereas the amount of surface area representing the upper vertical meridian is similar between the two groups.

**Table 1 | Mean surface area and coefficient of variation (CV) for early visual field maps**

|          | V1                 |      | V2                 |      | V3                 |      |
|----------|--------------------|------|--------------------|------|--------------------|------|
|          | Mean               | CV   | Mean               | CV   | Mean               | CV   |
| Adults   | 2129 mm²           | 0.21 | 2067 mm²           | 0.19 | 1680 mm²           | 0.16 |
| Children | 2013 mm²           | 0.27 | 1992 mm²           | 0.21 | 1603 mm²           | 0.18 |

The mean surface area is calculated as the combined left and right hemispheres of the map defined from 0 to 7° eccentricity. The CV reflects the variability of surface area of a field map relative to its mean and is calculated as the standard deviation of surface area/mean surface area.

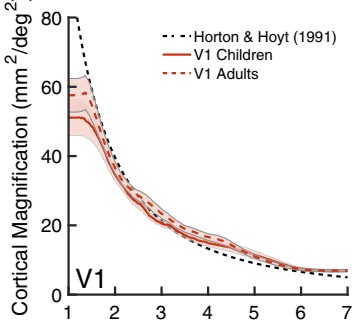
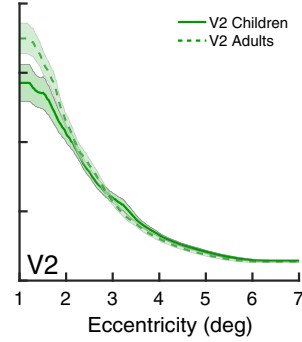
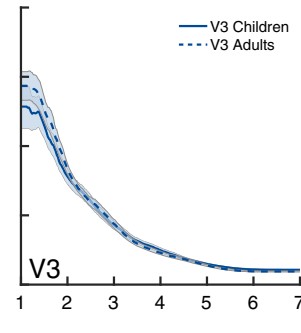

**Fig. 4 | Group-level areal cortical magnification as a function of eccentricity.** Adults (*n* = 24) are plotted as the dotted colored line and children (*n* = 25) are plotted in the solid-colored line. The black dashed line in the V1 panel represents the cortical magnification function of V1 reported by Horton and Hoyt (1991)[30]. The colored lines represent the group mean and shaded error bars are ±1 standard error of the mean (SEM). Source data are provided as a Source Data file.

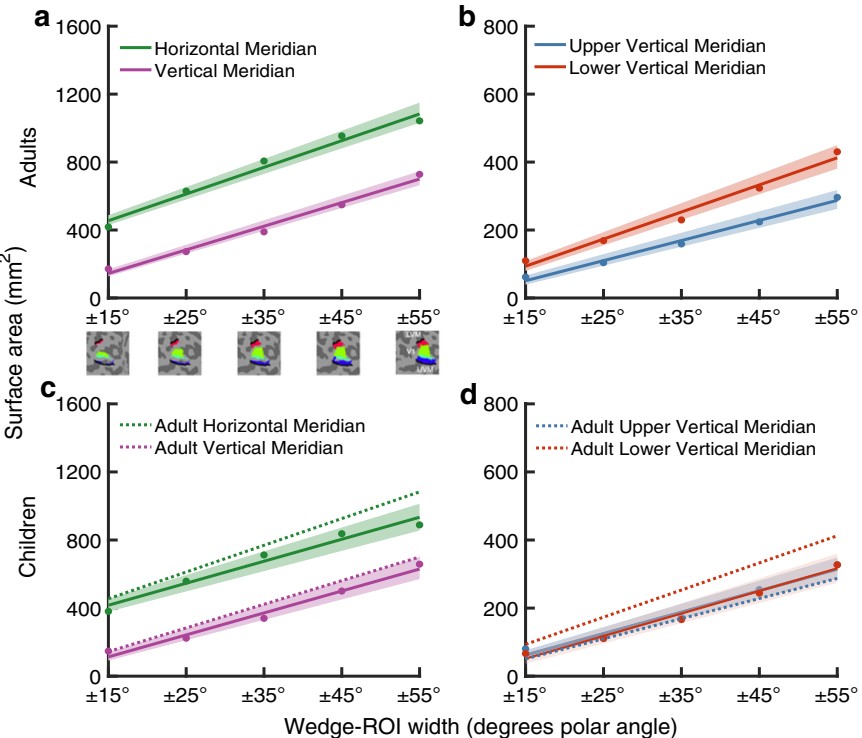

**Fig. 5 | Polar angle asymmetries in V1 surface area for adults and children.** **a–c** V1 surface area (mm²) measurements for wedges centered on the horizontal and vertical meridians, plotted as a function of wedge-width for adults ($n = 24$) and children ($n = 25$). Strip under (**a**) shows wedge-ROIs on the cortical surface. **b–d** V1 surface area measurements for wedges centered on the upper and lower vertical meridians, plotted as a function of wedge-width, for adults and children.

Colored lines represent the average of 10,000 bootstrapped linear fits to the data. The colored data points represent the bootstrapped mean at each wedge-ROI width. The shaded error bar represents the 68% bootstrapped confidence interval of a linear fit to the data. The dotted lines in (**c**) and (**d**) are the adult surface area measurements replotted for comparison to children. Source data are provided as a Source Data file.

The difference in the vertical meridian representation between adults and children was evident in the retinotopic maps of individual participants. In Fig. 6 we visualize polar angle maps of all children and one adult on inflated native cortical surfaces of the left hemisphere, zoomed and angled to show the boundaries of V1 that represent the lower and upper vertical meridians. In the example adult map, there is a thick red stripe along the dorsal V1/V2 border, representing the surface area of the lower vertical meridian. Maps from the left hemisphere of adults are available in Supplementary Fig. 7. Conversely, children have a much thinner red stripe, indicating a narrower representation of the lower vertical meridian that encompasses less cortical surface area.

Next, we parameterized the cortical asymmetries to quantify the magnitude of the cortical HVA and VMA. Here, we used the V1 surface area measurements taken from the ±25° wedge-ROIs at each of the four polar angle meridians (Fig. 7a). For each participant, we calculated an asymmetry index for the HVA as:

$$\text{Cortical HVA} = \frac{(\text{horizontal surface area} - \text{vertical surface area})}{\text{mean}(\text{horizontal surface area}, \text{vertical surface area})} \times 100 \quad (1)$$

A cortical HVA index of 0 indicates no difference in surface area between the horizontal and vertical meridian. As the asymmetry between the horizontal and vertical meridians increases, so does the magnitude of the cortical HVA.

Likewise, for each participant, we calculated an asymmetry index for the cortical VMA as:

$$\text{Cortical VMA} = \frac{(\text{lower vertical surface area} - \text{upper vertical surface area})}{\text{mean}(\text{lower vertical surface area}, \text{upper vertical surface area})} \times 100 \quad (2)$$

A cortical VMA index of 0 indicates no difference in surface area between the lower and upper vertical meridian. As the asymmetry between the lower and upper vertical meridian increases, so does the magnitude of the cortical VMA. A negative VMA index indicates an inverted VMA; more surface area for the upper than lower vertical meridian. These data met the assumption of normality and homogeneity of variance.

The magnitude of the cortical HVA was similar for children and adults, around 80 for both ($t(47) = -0.743$, $p = 0.461$, $d = 0.21$, $CI^{95} = [-27.02, 12.44]$, two-tailed independent samples t-test; Fig. 7b). However, the magnitude of the cortical VMA differed between children and adults ($t(47) = 3.631$, $p = 0.001$, $d = 1.04$, $CI^{95} = [21.41, 74.59]$, two-tailed independent samples t-test; Fig. 7c); children had a VMA close to 0, whereas for adults it was 52. In a supplementary analysis in children, we found no systematic relation between age and the magnitude of the HVA or VMA (Supplementary Fig. 9).

The pattern of results for the cortical HVA was consistent across individual children and adults (Fig. 7d); every individual –25 of 25 children and 24 of 24 adults– showed greater surface area for the horizontal than vertical meridian. Similarly, the pattern of results for the cortical VMA was consistent across individual adults; most showed greater surface area for the lower than upper vertical meridian (Fig. 7e). For children, individual data were close to the identity line (Fig. 7e).

Finally, to ensure that our results were not dependent on arbitrary choices of vertex selection, we computed the HVA and VMA using the ±25° polar angle wedge and six different eccentricity ranges (Supplementary Figs. 10 and 11). For all eccentricity ranges there was a robust HVA in adults and children and the VMA was always greater in adults than children.

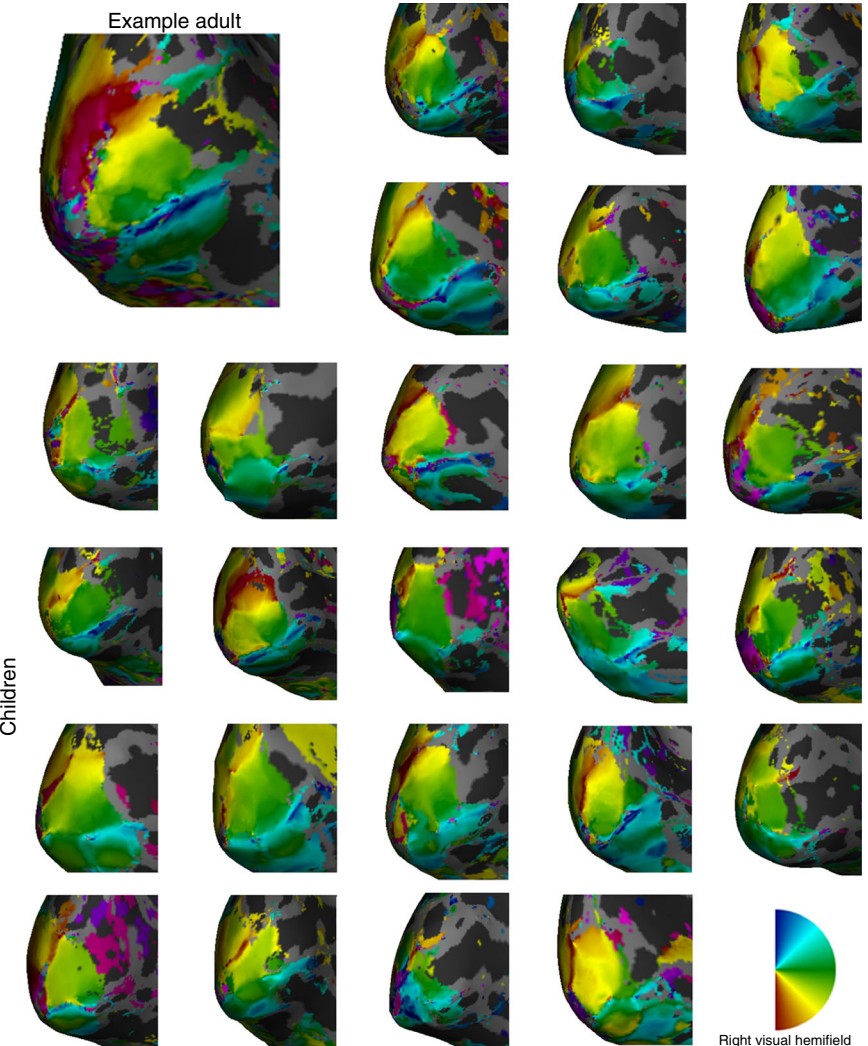

**Fig. 6 | Left hemisphere polar angle maps for an example adult and all 25 children.** Polar angle maps projected onto the left hemisphere of the inflated native surfaces angled to show the V1 representation of the right visual hemifield. The larger top left mesh shows an example adult. The other meshes come from the 25 children. Red colors correspond to the lower vertical meridian, blue to the upper vertical meridian, and green data to the horizontal meridian. Insert shows right hemifield polar angle map legend. Right hemisphere data for all children are available in Supplementary Fig. 8.

## Additional analyses to rule out motion and signal-to-noise as explanations of our findings

Within-scan motion presents the largest source of noise during fMRI[35] and children tend to move more than adults in the scanner[36,37]. Indeed, there was more within-scan motion for children than adults (median frames exceeding a motion threshold of 0.5 mm = 10.5/384 for children, 1/384 for adults; Supplementary Fig. 12). Although we do not see a plausible link between head motion and map size estimates, we nonetheless conducted several analyses to rule out the possibility that group differences in motion explained any of our main findings (Supplementary Note 1). Overall, participant motion did not impact the surface area measurements of maps or the magnitude of the cortical asymmetries for adults or children. These results indicate that the lack of a cortical VMA in children was not due to group differences in within-scan motion.

We also assessed whether asymmetries in V1 surface area were related to asymmetries in SNR. We tested this by computing VMA and HVA asymmetry indices in units of pRF variance explained ($R^2$) for each observer and compared these to VMA and HVA asymmetry indices in units of surface area ($mm^2$) for the same observers. There were no systematic relations across the two asymmetries and the two groups (Supplementary Fig. 13). These analyses indicate that neither the extent of the asymmetries in individual observers nor the difference in VMA between adults and children were an artifact of group-level variation in SNR.

## Discussion

We quantified and compared measurements of the cortical representation of the visual field between adults and children: retinotopic map surface area, cortical magnification as a function of eccentricity, and polar angle asymmetries in surface area. As expected, in V1, V2, and V3, both total surface area and cortical magnification as a function of eccentricity were similar between adults and children[1,3]. We then investigated a new domain of comparison: V1 surface area as a function of polar angle. First, these data revealed a commonality between adults and children – both showed much greater V1 surface area representing the horizontal than the vertical meridian of the visual field. Second, these data revealed a striking difference – adults had greater V1 surface area representing the lower than upper vertical meridian of the visual field, whereas children did not. This pronounced difference in the organization of V1 indicates that primary sensory cortices continue to develop beyond childhood. This late-stage change is surprising given that many properties of primary sensory and motor areas develop and mature early in life[1,2,12–14,38].

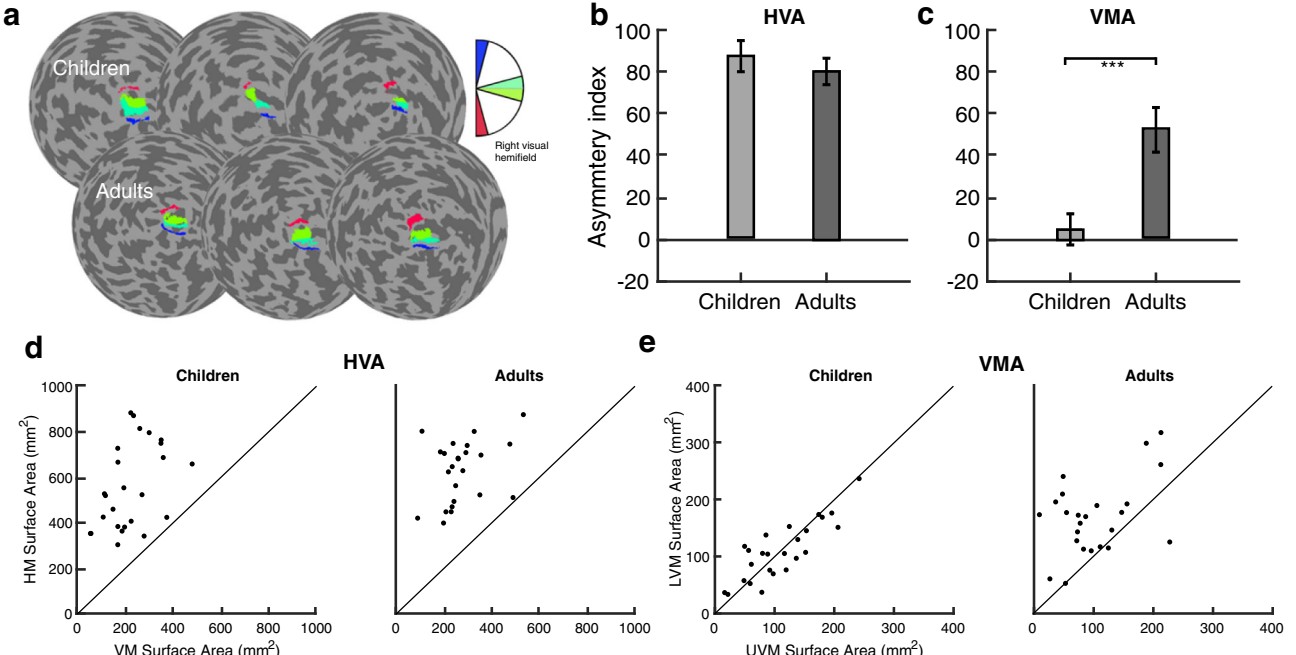

**Fig. 7 | Cortical HVA and VMA measurements from ± 25° wedge-ROIs, derived from pRF fits, centered on the four polar angle meridians. a** Examples of 25° wedge-ROIs overlaid on flatmaps of the left hemisphere cortical surface for 3 children and 3 adults. The colored surface areas on the cortical flatmap represent the corresponding regions of visual space in the inset legend. **b, c** Magnitudes of the Horizontal-Vertical Anisotropy (HVA) and Vertical-Meridian Asymmetry (VMA) index for children ($n = 25$) and adults ($n = 24$). Error bars indicate ±1 SEM. ***$p = 0.001$; two-tailed independent samples $t$-test. The individual surface area data plotted in panels **d** and **e** are used to calculate the HVA and VMA index in panels **b** and **c**. **d** Scatterplots of the HVA for children and adults, with data above the diagonal reference line indicating more surface area the horizontal than vertical meridian. **e** Scatterplots of the VMA for children ($n = 25$) and adults ($n = 24$), with data above the diagonal identity line indicating more surface area for the lower than upper vertical meridian, and vice versa. Source data are provided as a Source Data file.

## Cortical surface area as a dependent measure

There are many ways to compare visual cortex between groups, such as measures of BOLD amplitude, receptive field size, cortical thickness, and cortical surface area. Our primary dependent variable was cortical surface area –surface area of retinotopic maps, surface area as a function of eccentricity (cortical magnification), and the surface area of visual field-defined wedge-ROIs centered at the polar angle meridians. We focused on surface area for several reasons. First, surface area is an anatomical property, and its measure is robust to variation in experimental methods. In contrast, BOLD amplitude depends upon many unwanted factors such as field strength, pulse sequence, subject alertness, and properties of the cortex that are unrelated to neural signaling[39]. Likewise, estimates of pRF size are subject to noise and vary with the analysis algorithm[40], acquisition method[23], and attentional state of the participants[41]. Between-group variation in measurements of cortical thickness can be confounded by variation in myelination: comparisons showing differences in cortical thickness can either be due to true differences in thickness or to differences in myelination, such that segmentation algorithms misassign voxels in the deeper layers of gray matter as white matter[42]. This ambiguity may explain why estimates of thinner cortex are linked to sometimes better[43,44] and sometimes worse[45,46] visual performance. In contrast, greater surface area is linked to greater neural count[47], which is an important indicator of processing capacity. Greater surface area has been linked to better visual performance on a range of tasks[24,45,48–51]. For the sake of completeness, we also reported measures of cortical thickness and pRF size.

## The overall surface area of V1, V2, and V3 is stable between child- and adulthood

There was a close match between adults and children in: retinotopic map surface area, the degree of variability in map surface area, and

total surface area of the cortex. This is consistent with reports showing that the surface areas of V1, V2, and V3 are similar between adults and children[1] and that the total surface area of the cortical sheet is adult-like by age 10[52–54]. Notably, the coefficients of variation reported for V1, V2, and V3 were similar to those reported from adult data from the HCP dataset[28], all around 0.2. A coefficient of variation of 0.2 predicts a 2-fold range in map surface area when comparing the largest to smallest maps in sample sizes comparable to ours (20–30 participants, assuming approximately normal distributions), just as we found. A 2-fold range in V1 surface area has also been identified in infants under 5 years[38]. The substantial variability observed in the V1 surface area of infants, children, and adults is consistent with variability in other structures in the visual system. Cone density varies about 3-fold across individuals[55] and there is substantial variation in the size of the LGN and optic tract[56,57]. An intriguing possibility is that these structures co-vary, such that an individual with high cone density will also tend to have a large optic tract, LGN, and V1. Some older, post-mortem work[57], as well as recent in vivo measurements[56,58,59] suggest that this might be the case.

## Cortical magnification as a function of eccentricity is stable between child- and adulthood

Cortical magnification as a function of eccentricity in V1, V2, and V3 was similar between adults and children, consistent with prior neuroimaging reports[1,3]. fMRI work shows that eccentricity-dependent properties of adult V1 also exist in infant and child V1: for infants as young as 5 months, the spatial frequency tuning of retinotopic maps decreases with increasing eccentricity[38] and children as young as 5 years have adult-like pRF sizes that become larger with increasing eccentricity[1,2]. It is likely that the eccentric representation of the visual field is specified prenatally and fine-tuned by a visual experience early in life[38,60–62].

Surprisingly, however, foveal cone density increases into adolescence as cells migrate and pack more densely towards the fovea[63,64]. This could have some effect at the level of the cortex. There is a tendency in our cortical magnification data to see slightly greater magnification near the foveal representation of the visual field in adults than in children. This suggests that adults also have more cortical tissue representing the fovea than children, especially in V2 and V3. One possibility is that this effect is more apparent in V2 and V3 as these maps have greater cortical magnification of the fovea than V1[34]. Cortical mapping of the fovea is difficult[23,29,34] and retinotopic methods focusing on precisely and accurately mapping the foveal confluence may find that the development of this region is matched to retinal development, which reaches maturity by around 15 years of age[63,64].

### Cortical magnification as a function of polar angle changes between child- and adulthood

We identified a cortical HVA in adults, consistent with previous studies[23–26], and in children. Both groups had roughly twice as much V1 surface area representing the horizontal than vertical meridian of the visual field. These two groups are now the fifth and sixth to show a cortical HVA[23–26]. This finding is consistent with the notion that targeted fMRI measurements made with high signal-to-noise are highly replicable across groups with typical sample sizes (20–30 participants)[23,65].

We identified a prominent dissimilarity in the cortical VMA between adults and children. Children had no difference in the amount of V1 surface area representing the lower and upper vertical meridians of the visual field. Conversely, adults had more V1 surface area representing the lower than upper vertical meridian, consistent with prior work[23–26]. These findings parallel recent psychophysical findings showing that children have an HVA but no VMA in visual performance[15], whereas adults have both[15–22].

Nonetheless, the fMRI data do not perfectly match the psychophysical data. One discrepancy is that the magnitude of the behavioral HVA is smaller in children than in adults[15], whereas the magnitude of the cortical HVA is the same for both children and adults. Why might this discrepancy arise? There may be developmental changes outside V1 or changes in neural tuning that contribute to the larger behavioral HVA in adults that are not reflected by surface area measurements. A second discrepancy is that in adults, the magnitude of the behavioral HVA and VMA increase with eccentricity[18,20,66], whereas the cortical HVA and VMA measured here do not[25]. Importantly, the cortical and behavioral measurements come from different participants and are subject to natural individual variation as well as measurement noise. Hence, we remain agnostic as to whether the apparent discrepancies would hold when both measures are made in the same participants. To better understand these discrepancies and provide a link between cortical measures and visual performance at different locations in the visual field (both polar angle and eccentricity), we will measure both neural properties of V1 and visual performance in the same participants and use these measurements to develop a computational model that explicitly links differences in the distribution of V1 properties to visual performance throughout the visual field.

There is a correspondence between the cortical data reported here and prior behavioral data[15] at the group level: adults have a cortical and behavioral VMA, whereas children show neither. This contrasts with the effects at the individual level. We previously measured adults with both psychophysics and fMRI and found no correlation between the magnitude of the behavioral and cortical VMA[24]. We suspect that there are two reasons we find a positive relation measured at the group level but not the individual level. First, the group analysis benefits from averaging across many participants. Second, the variation of VMA magnitude is greater when including children than in an analysis that includes adults only.

Finally, our study is cross-sectional rather than longitudinal. As is always the case for cross-sectional studies, it is logically possible that the differences between groups could be cohort rather than age effects. However, we see no plausible explanation for a systematic cohort effect. The cortical VMA[23–26] and the psychophysical VMA[18,21,22,67,68] have been found in all adult populations tested. Moreover, the acquisition and analysis methods for children and adults were identical, ruling out methodological differences between the groups. While a longitudinal study spanning a decade or more could make the definitive case that the VMA develops between child- and adulthood, nonetheless, the developmental change in the V1 representation of the vertical meridian is the most plausible account of our data.

### Biological underpinnings: Changes in receptive field coverage and cortical geometry

The cortical VMA is a contrast between the V1 representation of the lower and upper portions of the vertical meridian of the visual field; in principle, the VMA could arise (between child- and adulthood) from an increase in the lower or a decrease in the upper vertical meridian representation of the visual field. Our data show that the cortical VMA in adults reflects an increase in the surface area representing the lower vertical meridian. We propose two possible neural underpinnings of this finding: changes in the spatial tuning of V1 receptive fields or local changes in properties of V1 tissue along the lower vertical meridian.

The VMA might emerge in adults due to remapping of the receptive fields along the lower vertical meridian. From childhood to adulthood, the receptive fields of V1 neurons near the lower vertical meridian could migrate closer to the lower vertical meridian –just as photoreceptors migrate closer to the fovea[63,64]. This would effectively increase receptive field coverage along the lower vertical meridian relative to the upper vertical meridian, resulting in a cortical VMA in adults. Prior work shows that pRFs remap across development in lateral visual field maps; from childhood to adulthood, lateral occipital (LO) and temporal occipital (TO) pRF coverage increases in fovea and periphery, respectively[69], and these changes result in an increase in the spatial precision of the pRFs in these regions[69]. Similarly, pRF remapping has been reported in dorsal maps, where pRF coverage becomes increasingly dense in the periphery between childhood and adulthood. For example, pRF sizes in IPS1 are larger and more eccentric in adults than children[70]. Further, and as noted above, there is evidence of cone migration through adolescence[63,64] and such changes in cone position could impact the receptive field properties of downstream visual neurons.

The explanation above posits a change in receptive fields rather than a change in cortical geometry. An alternative is that the amount of V1 surface representing the lower vertical meridian could increase due to changes in other tissue properties, such as dendritic spine density, glial cell properties, myelination, and remodeling of the extracellular matrix. Neuroimaging studies have reported activation-dependent changes in tissue properties in adults undergoing skill learning and perceptual learning, suggesting that the structure of the human brain can show long-term plasticity in response to environmental demands. For example, work using diffusion tensor imaging (DTI) has provided evidence for microstructural changes in the hippocampus after a spatial learning and memory task, likely due to reshaping of glial processes and the strengthening of dendritic spines[71]. Likewise, long-term training on a texture discrimination task has been associated with an increase in fractional anisotropy along the inferior longitudinal fasciculus, connecting V1 to anterior regions of the brain[72]. Similarly, fMRI work has shown that training in juggling is linked to transient increases in gray matter of motion-sensitive visual map hMT+[73] and faster phonetic learners have more white matter connections in parietal regions, indicating greater cortical myelination and thus more efficient processing[74]. Further, professional musicians have greater gray matter volume in motor, auditory, and visual regions than non-musicians[75]. None of these cases require the growth of new neurons, nor dramatic changes in neural tuning. Instead, these findings likely

reflect changes in the microstructural properties in which neural circuits are embedded. A similar effect might explain our data. Although it is not yet known what task demands, if any, specifically require greater use of the lower than the upper vertical meridian (rather than the lower vs the upper visual field) in adults than in children, we hypothesize that a change in input or behavior could result in activation-dependent changes in V1 tissue properties at cortical regions encoding the lower vertical meridian, manifesting as a change in the amount of V1 surface area devoted to this region of the visual field. Longitudinal data comparing the same individuals between child- and adulthood could distinguish between the two explanations by assessing whether tissue structure and cortical geometry change, or whether receptive fields change, or both.

## Ecological considerations

Children showed adult-like measurements of V1 surface area representing the horizontal meridian of the visual field. The finding that the cortical and behavioral HVA emerge early in development parallels the finding that the HVA is found early in the visual pathways (e.g., the density of cone photoreceptors[55,76]). Because of the importance of information along the horizon, many terrestrial animals have horizontal retinal streaks rather than circular foveae[77–79]; the human HVA may reflect similar environmental factors. Given the possible ecological advantage of a perceptual HVA for both children and adults, it may be that the horizontal representation of the visual field in V1 is broadly specified in-utero and fine-tuned during early years of life, similar to how eccentricity representations are thought to be a part of an innate cortical blueprint in human and non-human primates[38,61]. However, the specific image properties that might be important along the horizontal meridian (and their relevance to behavior) are yet to be elucidated.

In contrast, children did not have a cortical VMA, matching their lack of a behavioral VMA. A possible ecological account of this phenomenon would propose that a lower visual field specialization emerges with age due to neural adaptation to regularities of the visual environment. As discussed in prior work[15], given children's height and behavior, a large portion of their perceptual world is within the upper portion of the visual field. Thus, a VMA would be detrimental to their interaction with their visual environment. As children grow to become adults and their height increases, their visual environment consists of more important events and actions below fixation[80] –for example, visuomotor manipulation such as reaching and tool use[81–83]. This is congruent with evidence that the behavioral VMA emerges in late adolescence[84]. We plan to investigate whether the cortical VMA emerges at a similar period by quantifying the cortical polar angle asymmetries in adolescents, allowing us to uncover whether the emergence of the behavioral VMA is linked to the emergence of the cortical VMA.

However, it is not yet known what task demands, if any, specifically require greater use of the lower vertical meridian (rather than lower visual field) in adults than in children. Polar angle asymmetries in visual performance[16,18,85] and cortical surface area[23,25] are most pronounced at the cardinal meridians and gradually diminish to the point that they are often no longer present at intercardinal locations. Thus, any specialization must be primarily driven by the vertical meridian itself.

To conclude, these data show that many aspects of early visual field maps (V1-V3) are mature by early childhood –map surface area, cortical magnification as a function of eccentricity, and greater surface area representing the horizontal than vertical meridian of the visual field. However, children, unlike adults, do not have a cortical asymmetry in the amount of V1 surface area representing the upper and lower vertical meridian. These findings reveal a large, late-stage change in how V1 samples the visual field that parallels the presence of a vertical meridian asymmetry in visual perception by adulthood.

## Methods

### Participants

Raw retinotopy data (25 children and 26 adults) were obtained from a previous fMRI study investigating receptive field development in children[2]. 2 adults were not included in the analyses as we could not reliably identify their V1 boundaries (Supplementary Fig. 14). Thus, data from 25 children (ages 5–12 years old; 11 males, 14 females, mean = 8.6 years) and 24 adults (ages 22–27 years old, 13 males, 11 females, mean = 23.8 years) were included in the study. Participant sex was assigned based on self-report. All participants had normal or corrected-to-normal vision. Written informed consent was obtained for all participants; all parents provided written consent for their child to participate in the study and children provided written assent. The retinotopy experiment was conducted in accordance with the Declaration of Helsinki and was approved by the Institutional Review Board of Stanford University. Data used in this study are publicly available on OpenNeuro as the Stanford Child and Adult Checkerboard Retinotopy Dataset.

### Retinotopic mapping experiment

For each adult, data were collected across two scan sessions –one anatomical and one functional session. For each child, data were collected over three sessions –one mock scanner training, one anatomical, and one functional session– across the span of a few months. First, to acclimatize children to the MRI scanner and to practice lying still, they were required to complete training in a mock scanner. During the training, the child participant viewed a movie (~15 min) in the scanner and received live feedback of head motion. Next, each participant completed an MRI session in which a full brain anatomical image was obtained. Finally, each participant completed an fMRI session in which they completed four runs of a retinotopic mapping experiment to measure population receptive field (pRF) parameters across visual cortex.

The pRF stimulus and the MRI and fMRI acquisition parameters were identical to prior work[2]. However, the data were reanalyzed, thus the MRI and fMRI preprocessing and the implementation of the pRF model, differ.

### fMRI stimulus display

Participants viewed the pRF stimulus from within the scanner bore; the stimulus image was projected onto an acrylic screen using a rear-projection LCD projector (Eiki LC-WUL100L Projector). The projected image had a resolution of 1024 × 768. Participants viewed the screen from a distance of 265 cm inside the scanner bore using an angled mirror that was mounted on the head coil.

### pRF stimulus

Retinotopic maps were measured using pRF modeling[86]. The pRF stimulus was generated using MATLAB and was projected onto the fMRI stimulus display in the scanner bore using the Psychophysics Toolbox v3[87]. The pRF stimulus consisted of 100% contrast black and white checkerboard patterns that were presented within a bar aperture that swept across the screen for the duration of each scan, as in Dumoulin and Wandell (2008)[86]. The checkerboard pattern was windowed within a circular aperture that had a radius of 7°. The checkerboard pattern was revealed through a bar aperture (width = 3°, length = 14°) that swept across the screen. Each step was synchronized to the fMRI image acquisition (TR = 2 s). There were 8 sweeps in total and each sweep began at the edge of the circular aperture. Horizontal and vertical sweeps covered the entire diameter of the circular aperture, whereas diagonal sweeps only traversed half the circular aperture, with the second half of the diagonal sweep being replaced with a blank gray screen. The full stimulus run lasted 3 min and 24 s. The stimulus drift velocity was 0.30° per second. The stimulus image updated 2 times per second without intermediate blanks.

During each pRF stimulus run, participants were instructed to maintain fixation on a small spaceship (~0.5° in size) located in the center of the display that acted as a fixation stimulus throughout the entire scan. Participants performed a fixation task in which they were required to respond, via a button box, when the spaceship changed color. An Eyelink 1000 was used to monitor and record fixation within the scanner. Eye tracking data were obtained for 25 children and 6 adults. Data could not be obtained from most adults due to participant head size and time constraints. Any scans in which children broke fixation more than twice were discarded and scanning was restarted. The retained data showed small numbers of fixation breaks in both groups, an average of 0.5 fixation breaks per group in adults and 1.5 in children. Fixation task accuracy was above 98% in both groups. See Gomez et al. (2018)[2] Supplementary Fig. 1 for detailed eye tracking results.

## Anatomical and functional data acquisition

Anatomical and functional data were acquired on a 3-Tesla GE Discovery MR750 scanner (GE Medical Systems) at the Center for Cognitive Neurobiological Imaging at Stanford University.

Quantitative magnetic resonance imaging (qMRI) full-brain anatomical measurements were obtained using the protocols in Mezer et al. (2013)[88] and with a phase-array 32-channel head coil. T1 relaxation times were measured from four spoiled gradient echo (spoiled-GE) images (flip angles: 4°, 10°, 20°, 40°; TR: 14 ms; TE: 2.4 ms; voxel size: 0.8 mm × 0.8 mm × 1.0 mm). To correct field inhomogeneity, four additional spin inversion recovery (SEIR) images were collected (inversion times: 50, 400, 1200, 2400 ms; TR: 3 s; echo time set to minimum full and 2x acceleration; voxel size: 2 mm × 2 mm x 4 mm) with an echo planar imaging (EPI) read-out, a slab inversion pulse, and spatial fat suppression.

Four functional EPI images (96 volumes per image) were acquired for each participant using a 16-channel head coil and a multiband EPI sequence (TR, 2 s; TE, 30 ms; voxel size, 2.4 mm³ isotropic; multiband acceleration factor, 2, 28 slices) with slices aligned parallel to the parieto-occipital sulcus.

## Processing of anatomical data

qMRI anatomical measurements (i.e., the spoiled GE image and the SEIR images) were processed using mrQ software[88]. The mrQ pipeline corrects for radiofrequency coil bias using the SERI images to produce accurate proton density and T1 fits across the brain. These fits were used to compute a T1w anatomical image for each participant.

Following this, fMRIPrep v.20.0.1[89,90] was used to process the anatomical and functional data. For each participant, the T1w anatomical image was corrected for intensity inhomogeneity and skull stripped. The anatomical image was then automatically segmented into cerebrospinal fluid, cortical white-matter, and cortical gray-matter using fast[91]. Cortical surfaces in the participant's native space (called fsnative in FreeSurfer) at the midgray, pial, and white matter depth were reconstructed using FreeSurfer's recon-all[92].

## Processing of functional data

The following processing was performed on each participant's functional data. First, a reference volume (and skull stripped version) was generated using custom methodology of fMRIprep. An estimated distortion of the B0-non uniformity map was used to generate a corrected functional reference image; this reference image was co-registered to the anatomical image using six degrees of freedom. Following this, head-motion parameters with respect to the functional reference were estimated before any spatiotemporal filtering. Each functional image was then slice-time corrected. All slices were realigned to the middle of each TR. These slice-time corrected data were then resampled to the T1w anatomical space via a one-shot interpolation consisting of all the pertinent transformations. These preprocessed time-series data were then resampled to the fsnative surface for each participant, by averaging across the cortical ribbon. For each participant, this processing was completed for each of the four functional scans.

## Within-scan motion: Calculating framewise displacement

Framewise displacement (FD) was calculated as a quality metric for motion artifacts based on the framewise differences of the 3D translation (mm) and rotation (radians) estimates of motion during the scan. First, we took the absolute framewise difference of the translation and rotation metrics across three dimensions:

$$\Delta d_{tx} = d_{(t-1)x} - d_{tx} \tag{3}$$

Where d represents the translational motion estimate, t represents time, and x represents the slice dimension.

To combine the rotational motion with translational motion in mm units, we estimated the arc length displacement of the rotational motion metric from the origin by approximating the distance between the center of the head and the cerebral cortex with a radius of 50 mm[93].

$$\Delta\alpha_t = 50 \cdot (\pi/180) \cdot (\alpha_{(t-1)} - \alpha_t) \tag{4}$$

Where $\alpha$ represents the rotational motion estimate, and t represents time.

Finally, we calculated the FD as the sum of the framewise difference of the translational and rotational metrics across three dimensions:

$$FD_t = |\Delta d_{tx}| + |\Delta d_{ty}| + |\Delta d_{tz}| + |\Delta\alpha_t| + |\Delta\beta_t| + |\Delta\gamma_t| \tag{5}$$

For each participant, we counted the number of times the FD exceeded the threshold of 0.5 mm across scanning sessions (see fMRIprep documentation for the set threshold[89]).

## Implementing the pRF model on the cortical surface to compute retinotopic maps

The pRF model was implemented using vistasoft (Vista Lab, Stanford University). We used customized code to run the pRF model on the cortical surface. Here, a pRF was modeled as a circular 2D Gaussian which is parameterized by values at each vertex for x and y (specifying the center position of the 2D Gaussian in the visual field) and σ, the standard deviation of the 2D-Gaussian (specifying the size of the receptive field). This 2D Gaussian was multiplied (pointwise) by the stimulus contrast aperture and was then convolved with the canonical two-gramma hemodynamic response function (HRF) to predict the BOLD percent signal change (or BOLD signal). We parameterized the HRF using 5 values, describing a difference of two gamma functions[23,86,94–96].

For each participant, the time-series data were averaged across the four preprocessed functional images to create an average time-series. This average time-series was then transformed to a BOLD signal. For each participant, the pRF model was fit to the BOLD signal for each vertex on the fsnative surface (i.e., the native surface generated by Freesurfer). The pRF model finds the optimal pRF parameters for each vertex by minimizing the residual sum of squares between the predicted time-series and the BOLD signal. The pRF model was fit using a multi-stage coarse-to-fine approach[95]. This approach was designed to reduce the chance of the search algorithm getting stuck in a local rather than global optimal solution, and more generally to reduce the chance of finding a solution that fits the noise rather than the signal. First, the data were temporally decimated by a factor of two to remove high frequency noise. Next, the pRF parameters (x, y, σ) were fit using a brute force grid search. The results were then taken as the starting point of a second-stage search fit. The estimated pRF parameters were then held fixed and the HRF parameters were fit by a search by

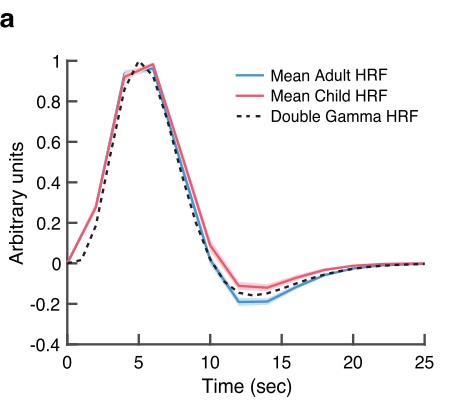

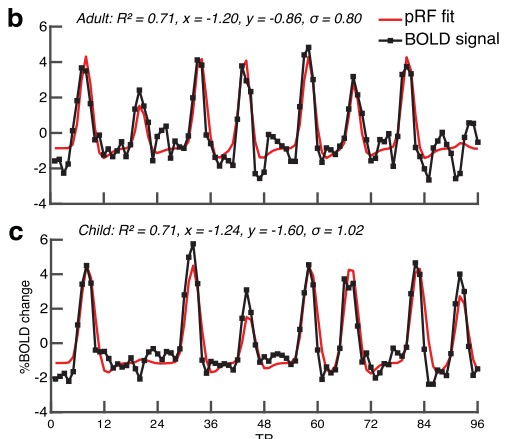

**Fig. 8 | Mean adult and child HRF and examples of the pRF model fit to the BOLD signal. a** The mean HRF for adults ($n = 24$; blue) and children ($n = 25$; red), and the double-gamma first-pass HRF (dashed black line). Shaded error bars are ± 1 SEM. **b**, **c** The BOLD signal (black) and pRF model fit (red) for a V1 vertex for an (**b**) adult and (**c**) child participant. The inset text lists the $R^2$ value which is the variance explained of the pRF model fit to the BOLD signal, the predicted $x$ and $y$ values representing the pRF center coordinates, and the predicted σ value, representing pRF size.

choosing parameters that minimize the squared error between the data and the prediction averaged across vertices. Finally, the HRF parameters were held fixed and the pRF parameters were refit to the data in a final search fit. See Fig. 8a for the mean estimated HRF for adults and children. Adults had a slightly stronger post-stimulus undershoot than children, whereas previous reports have shown the opposite[97]. See Fig. 8b, c for the pRF model fit to the BOLD signal of a V1 vertex for example adult and child participants.

### Defining V1, V2, and V3
V1, V2, and V3 were defined as regions-of-interest (ROIs) by hand (M.M.H and E.T) using *Neuropythy* v0.11.9 (https://github.com/noahbenson/neuropythy[98]). Each ROI was defined on flatmaps of the cortical surface and extended from 0°to 7° of eccentricity. The V1, V2, and V3 boundaries were defined as the center of the polar angle reversals that occurred at the vertical or horizontal meridians[23,86,99]. Only data with an $R^2 \geq 0.10$ were used to delineate the maps.

### Calculating cortical magnification as a function of eccentricity
For each observer, we calculated the cortical magnification function as a function of eccentricity for each of the V1, V2, and V3 ROIs using *Neuropythy* v0.11.9[98]. For each participant, and for a given ROI, we calculated cortical magnification $m(r)$, or mm$^2$ of cortex/degrees$^2$ of visual space, from 1° to 7° of eccentricity. We did so by finding the value Δ$r$, given that 20% of the *fsnative* vertices had an eccentricity between $r - \Delta r$ and $r + \Delta r$. The surface area (mm$^2$) of these vertices was summed and divided by the amount of visual space (deg$^2$) encapsulated by the eccentricity ring, which is defined as $r \pm \Delta r$. This effectively defines an annulus in the visual field and calculates the amount of cortical surface area representing this annulus. This results in a measurement of areal cortical magnification from 1 through to 7° of eccentricity.

### Calculating V1 surface area as a function of polar angle
For each observer, we calculated localized measurements of V1 surface area (mm$^2$) representing the polar angle meridians. We did so by defining wedge-ROIs that were centered on the polar angle meridians in the visual field. These wedge-ROIs were defined using *Neuropythy* v0.11.9 and custom MATLAB code. The specific implementation of this analysis is described below and in prior work[23,24] and is robust against variation in SNR while circumventing any discontinuities in surface area that would arise if we had defined angular regions of visual space using estimates of pRF centers alone[23,98]. However, this method relies

on pRF estimates to define wedge-ROI borders, and pRF estimates are robust against variation in BOLD amplitude[40,100].

In brief, the process was as follows. A wedge-ROI was centered on either the left or right horizontal meridian, the upper vertical meridian, or the lower vertical meridian. The polar angle width of the wedge-ROI varied, extending ± 15°, ± 25°, ± 35°, ± 45°, and ± 55° in angle from the respective meridian. Each wedge-ROI extended from 1 to 7° of eccentricity. Unlike the 0°–7° limit we used for defining V1-V3, we excluded the central 1° from the wedge-ROI because polar angle representations can be noisy in the fovea[23,29,34]. Each wedge-ROI was used as a mask and was overlaid on a cortical surface area map. These maps specify the cortical surface area (in mm$^2$) of each vertex on the *fsaverage* surface. The amount of cortical surface representing the polar angle meridians was calculated by summing the surface area of the vertices that fell within the wedge-ROI mask.

### Defining wedge-ROIs along the polar angle meridians
The following processes were completed for each observer using *Neuropythy* v0.11.9[98]. First, the shortest distance on the *fsnative* surface between each pair of vertices and a polar angle meridian was calculated. The horizontal, upper, and lower vertical meridians in V1 were defined using manually defined line-ROIs that were drawn upon the polar angle pRF data. The vertical meridian line-ROI followed the trace of the defined V1 ROI. The line-ROIs were used to calculate three cortical distance maps per hemisphere: one for the upper vertical meridian, one for the lower vertical meridian, and one for the horizontal meridian. The cortical distance maps specified the distance (in mm) of each vertex from the meridian (i.e., there was one cortical distance map per meridian per participant). Thus, the vertices along the meridian itself had a distance value of 0 mm. This process was repeated for the left and right hemispheres of V1, so that the upper and lower vertical wedge-ROIs would span the left and right visual hemifield.

The V1 map was then divided into 7 log spaced eccentricity bands between 1° and 7° of eccentricity in the visual field. This ensured that the eccentricity bands were roughly equally spaced in V1. The eccentricity bands were used to calculate sub-wedge-ROIs. These would be later combined to form a full wedge-ROI. Thus, each wedge-ROI is formed of multiple smaller sub-wedge-ROIs that represent some eccentricity portion of the full wedge). The eccentricity bands were defined using retinotopic maps generated by Bayesian inference[98] to ensure that each eccentricity band was a continuous region of visual space. These Bayesian inference maps combine each participant's vertex-wise pRF estimates with a previously defined retinotopic

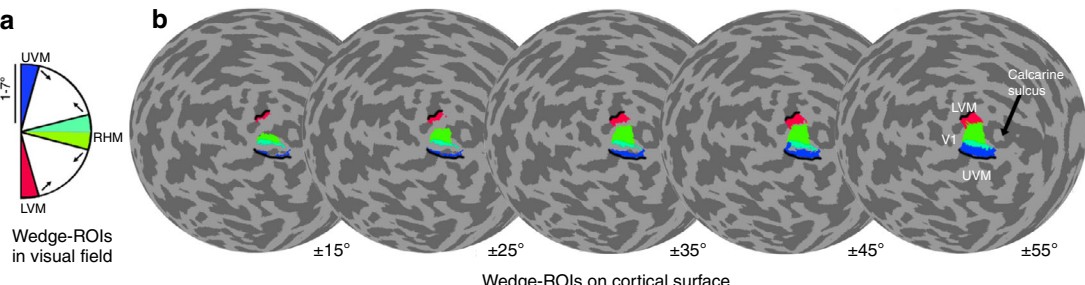

**Fig. 9 | Wedge-ROI masks on the left cortical hemisphere of an example participant. a** PRF centers are used to compute a series of masks that are used to define wedge-ROIs in the visual field. **b** The wedge-ROI masks are overlaid on the cortical surface. The mean surface area is computed for each wedge-ROI. This is repeated for each wedge-ROI width (15°–55°). As expected, the colored wedge-ROIs become larger as wedge-width increases. Black lines represent the dorsal and ventral borders of V1/V2. Each sphere is the native surface that has been transformed to a flattened sphere with the occipital pole aligned to the center.

template to denoise the estimate of the visual field. The polar angle maps were cleaned using *Neuropythy*[98], by implementing an optimization algorithm on the pRF polar angle fits in V1 for each participant. This minimization aims to adjust the pRF centers of the vertices as little as possible to simultaneously enforce a smooth cortical magnification map and correct the field-sign values across V1. Both the Bayesian inference eccentricity maps and the cleaned polar angle maps were only used in the wedge-ROI analysis.

For each wedge-ROI, and within each eccentricity band, we used the cortical distance maps to compute the distance of an iso-angle line that represented the outer boundary of the wedge-ROI in the visual field. We measured wedge-ROIs of multiple widths. The iso-angle line fell 15°, 25°, 35°, 45°, and 55° of angle away from a meridian. This effectively forms wedge-ROIs in the visual field that extend 15°, 25°, 35°, 45°, and 55° from a meridian. Our goal was to calculate the distance (in mm of cortex) of this iso-angle line from a meridian. To do so, we used the cleaned pRF polar angle data and the cortical distance maps to calculate the average distance of the vertices in a region of V1 that fell $\pm 8°$ around the iso-angle line polar angle value and an $R^2 \geq 0.10$. The average distance of these vertices represents the distance on the cortex (in mm) of the iso-angle line from a meridian. This process was repeated for each eccentricity band.

For each eccentricity band we identified the vertices with a cortical distance value of 0 mm (i.e., those along a meridian, and thus the center boundary of the wedge-ROI) and with a distance pertaining to each iso-angle boundary (i.e., 15° through to 55°). This was repeated for each eccentricity band to generate sub-wedge-ROIs. These sub-wedge-ROIs were combined to create a full wedge-ROI. This process was repeated for each meridian and each hemisphere. Together, these steps identify the vertices that will form the center-boundary of the wedge-ROI (i.e., a polar angle meridian) and the vertices that form the outer-boundary of that wedge-ROI. The vertices within these boundaries approximately represent a continuous wedge in the visual field.

The final step was to use the full wedge-ROI as a mask on cortical surface area maps. These maps are generated for each observer and denote the surface area (in mm²) of each vertex on the *fsnative* surface. The wedge-ROI was overlaid on the cortical surface area map and the surface area of the vertices within the wedge-ROI were summed to calculate the surface area of the wedge-ROI. This was repeated for each meridian, and each hemisphere. See Fig. 9 for visualization of wedge-ROI masks overlaid on the cortical surface. To calculate the amount of surface area dedicated to processing the horizontal meridian, the surface area of the left and right horizontal wedge-ROIs were summed together. To calculate the amount of surface area dedicated to processing the upper vertical meridian, the surface area of the wedge-ROIs extending from the left and right sides of the upper vertical meridian (thus right and left hemisphere's of V1, respectively) were summed

together. Likewise, to calculate the surface area dedicated to processing the lower vertical meridian, the wedge-ROIs extending the left and right sides of the lower vertical meridian (again, the right and left hemispheres of V1, respectively) were summed together. The upper and lower vertical meridian wedge-ROIs were summed to calculate the surface area dedicated to processing the full vertical meridian. Importantly, whereas vertices with an $R^2 \geq 0.10$ were used to find the boundaries of these ROIs, all vertices, including those with an $R^2 \leq 0.10$ were used in the computation of the wedge-ROI surface area measurements.

### Midgray, pial and white matter surfaces
Cortical surface area was measured using cortical surface area maps that were generated, using FreeSurfer, at three levels of surface depth: midgray, pial, and white matter. All main analyses were conducted on the midgray surface of the cortex which falls in the middle-depth of the cortical sheet between the pial and white matter surfaces. Supplemental analyses were calculated on pial (upper) and white matter (lower) surfaces. The cortical surface area of the vertices within the three maps differ; this is because the surface area of gyri at the pial surface and the sulci at the white matter surface tend to be large, whereas the sulci at the pial surface and gyri at the white matter surface tend to be small.

### Reporting summary
Further information on research design is available in the Nature Portfolio Reporting Summary linked to this article.

## Data availability
Raw data used in this study were taken from previous work (Gomez et al. 2018, https://doi.org/10.1038/s41467-018-03166-3). Raw and preprocessed MRI and fMRI data, and pRF model fits that were generated in this study, are publicly available in BIDS format and have been deposited in OpenNeuro (accession code: ds004440) as the Stanford Child and Adult Checkerboard Retinotopy Dataset (https://openneuro.org/datasets/ds004440). Source data are provided with this paper.

## Code availability
Scripts used for data collection are available through the open source vistasoft library (https://github.com/vistalab/vistadisp) and code to generate key manuscript figures are available on the OSF repository (https://osf.io/2yqwe/).

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

## Acknowledgements

This work was funded by NIH grants R01-EY027401 to M.C. and J.W., P30-EY013079 to Tony Movshon (PI), M.C., and J.W. (module directors), R01-EY022318 and R01-EY023915 to K.G.-S., and Princeton Neuroscience Institute start-up funds to J.G. We extend our thanks to Eline Kupers for her assistance in organizing and sharing the raw MRI data from K.G.-S.'s lab and Michael Barnett for assisting in data collection.

## Author contributions

M.C. and J.W. conceived the experiments, J.G. and K.G.-S. provided the data, M.M.H., E.T., J.G., and J.W. analyzed the data, M.M.H., E.T., M.C., and J.W. wrote the paper, and all authors edited the paper. M.C. and J.W. jointly supervised the work.

## Competing interests

The authors declare no competing interests.
