## [Peer Review File · Nature Communications]

Comparing retinotopic maps of children and adults reveals a late-stage change in how V1 samples the visual fieldREVIEWER COMMENTS

Reviewer #1 (Remarks to the Author):

The ms reports very interesting data on the development of retinotopic asymmetry of primary visual cortex. In adult, in addition to horizontal and vertical meridian asymmetry, already well documented in the literature, are large, while the upper-lower vertical meridian asymmetry is small and difficult to document. These authors show that the vertical meridian asymmetry is not present in children 7-12 years old. Given the small effect the work is quite heroic. The presented data are convincing and clear: the polar maps are different between adults and children. However, I think that the result may be explained in different ways, and I hope that the authors could provide additional analysis that could dismiss these alternative explanations. I would have no hesitation in recommending publication given the implication of the results, if the author could provide this additional evidence.

1) The strongest evidence of the data are from the polar-map. This method relies on the strength of the BOLD responses. So if the lower field excitation produces a stronger BOLD inevitably the area associated with the stimulation will be larger (classical problem of circular statistic). The authors should demonstrate that the BOLD response of the lower and upper field have the same amplitude and the same S/N in children; and, crucially, also in adult (a clear problem that was not raised in the previous published paper);

2) The maps obtained from pRF with drift bars are less subject to the BOLD artefact. Unfortunately the authors did not attempt to generate an average pRF retinotopic map across participants to illustrate the map deformation. However, they evaluated the asymmetry by counting on individual participants the vertices associated with a particular orientation. The obtained results of fig S4 are not completely consistent with the results of the polar map, reinforcing the idea that BOLD amplitude may have played a role in estimating the asymmetry. Is the asymmetry difference significant in adults? The pRF model gives also pRF size, which may be different between upper and lower visual fields. These are important data that need to be presented.

3) The stronger evidence in support of the asymmetry may derive from anatomical measures, not subject to BOLD response amplitude. In many visual pathologies also in children V1 thickness can change dramatically compared with typical participants. Please report the average thickness for the upper and lower vertical meridia. In this case the BOLD defined Roi can be used, given that it is an independent measure.

4) 24 participants is a large sample to attempt correlation analysis. The asymmetry of the horizontal and vertical meridian and of the upper and lower vertical meridian varies greatly between participants and this allows a correlative analysis with psychophysical thresholds. I suppose that many participants are the same of those of the recent Current Biology paper. If not these measures are very easy to collect in children, even with web based procedures. A correlation with psychophysics would greatly strengthen the results.

5) Eye movements are never mentioned or discussed in the manuscript. I understand that not all scanners have the capability to control fixation but at least the issue should be raised. The larger and noisier foveal confluence could be due to increased frequency of saccades in children?

6) Could eye movement direction induce the vertical meridian asymmetry, endorsing a stronger influence of the motor system in the lower visual field representation in V1?

7) The fact that overall surface area of V1 and V2 are equal in children and adult, a greater representation of the lower visual field should move away from the fundus of V1 the representation of the horizontal meridian. Can please mark the fundus of calcarine sulcus on the retinotopic map?

8) 5 to 12 is a rather large age range. Do the older children show some anisotropy?

Minor

Line 72. Cortical magnification here needs to be specified. Overall the main point of the paper is to demonstrate a different cortical magnification for upper and lower visual field!

Fig 1 Please give the R^2 of the fit of the map. In adults the asymmetry is NOT very evident!!!

Fig3 A and B have the same y label, generating a lot of confusion.

175 "within an eccentricity ring" – please rephrase the unclear sentence

544 please report the stimulus drift velocity.

Concetta

Reviewer #2 (Remarks to the Author):

This paper builds on three recent sets of findings: 1) Children and adults show a visual sensitivity advantage for stimuli on the horizontal compared to the vertical meridian (the HVA). Adults also show an advantage for the lower over the upper vertical meridian, (the HVA), but -crucially - this is not seen for children (Carrasco et al., 2022 *Curr Biol*). 2) Individual differences in the adult HVA, but not the VMA correlate with corresponding anisotropies in cortical magnification (Himmelberg et al., 2022 *Nat Comms*). 3) The functional layout of early visual cortex in children appears adult like at an early age (e.g. Dekker et al., 2019 *Dev Cogn Neurosci*).

However, previous studies on the development of retinotopic maps in children have not considered polar angle anisotropies in cortical magnification (CMF). The current paper sets out to fill this gap using fMRI and population receptive field mapping. The authors indeed find clear HVA and VMA for V1 cortical magnification in adult V1, but only an HVA for children, matching recent findings in psychophysics.

This finding links between-subject variance in cortical magnification to that in visual sensitivity, which so far has not been possible for the VMA (Himmelberg et al., 2022). Additionally - and more importantly - it provides potential evidence for unexpectedly late large scale changes in the functional organization of early visual cortex. The latter has wide ranging implications for visual neuroscience and general mechanisms of cortical organization, rendering the finding appealing for a wide audience.

The hypotheses, experimental logic, writing and evidence are exceptionally clear. The methods appear solid and the authors took particular care to control for potential confounds by motion artefacts in children. However, I would like to suggest a few additional controls and know the authors' opinion on possible alternative interpretations of their findings.

MAJOR

-> AGE VS COHORT EFFECTS. Given you compare cross-sectional samples, could the findings be due to cohort effects rather than age? Specifically, children aged 5-12 today probably have had more (small) screen time than current 20-somethings had at the same age. Younger cohorts potentially also spent less time outdoors (as suggested by the myopia epidemic). Please discuss.

-> POTENTIAL GROUP DIFFERENCES IN SPATIAL VARIANCE OF SNR. I think it is necessary to control for group differences in location-varying SNR effects, such as those induced by draining veins (Winawer et al., 2010 *JoV*). Relative to adults, is there a dip in the R2 map of children around the lower vertical meridian?

-> POTENTIAL IMPACT OF PRIORS AND TEMPLATES ON RESULTS (lines 706ff). The potential impact of the Bayesian eccentricity template and polar angle cleaning algorithm seem important, esp. given templates probably are derived from adult data(?). To what degree do the results hinge on these processing steps? Does the general pattern of results replicate when only using the empirical data, e.g. by calculating the sum of cortical area for all vertices with pRFs falling into a given visual field ROI, regardless of map discontinuities (this may be improved by simple surface smoothing)? If this is too noisy, is there any other way of ensuring the pattern of results isn't imposed onto the data by algorithms imposing prior assumptions?

-> CHANGES IN STRUCTURE VS CHANGES IN pRF MAPPING ONTO STRUCTURE. The distinction discussed in lines 433ff seems relevant. Isn't it possible to juxtapose the two hypotheses with the data you have at hand? If the surface area of the upper bank of the calcarine expands disproportionately in adults (the 'structure' hypothesis), this should become apparent by comparing the surface area of the upper and lower bank between groups, only using the meridian lines to define them (comparing whole quarter fields rather than polar wedges). However, if the structure stays the same and only pRFs migrate within the V1 map (the 'mapping' hypothesis), such an analysis should not yield any group differences. But maybe I'm missing something?

MINOR

-> Figures S7 and S8 suggests that the adult VMA in V1 is entirely driven by the close parafoveal range below 2 or 3 degrees of eccentricity. I think this finding deserves more attention and should be discussed, maybe in the context of retinal developments in line 400ff. Is this a real effect or more likely an artefact of data attrition for higher eccentricity bands? What's the evidence for the perceptual VMA at different eccentricity bands?

-> Given the broad age range among children, is there a trend for the VMA emerging in older children? It would be nice to see a scatter plot of VMA vs age.

-> Please show some example time-series and fits from adult and children's vertices (e.g. in Figure 1)

-> lines 130ff: the high degree of similarity between children and adults is very apparent in Figure 3. However, I don't think a ' $p > 0.5$ ' is informative or appropriate in this context. If using tools from inference statistics, I suggest to report the confidence interval for a mean difference between groups instead to show which effect sizes can be excluded as unlikely (mean differences probably best expressed in % of adult total surface area).

-> Lines 162ff I appreciate the efforts to control for potential motion confounds. However, I don't quite get the point of this analysis. Is there any reasonable expectation that surface area estimates would correlate with motion? If so, please briefly explain the reasoning. Similarly for lines 313ff. Also, it would be good to give the reader the descriptive stats from Figure S1 here already (median number of frames with excessive motion 1 for adults, 10 for children). Regarding the latter, I think the methods section fails to state the number of volumes (only four runs mentioned, unless I missed it). Please add for context.

-> what central tendency do the lines in Figure 4 show? Means? Medians? Please add

-> It may be worth explicitly discussing possible reasons for the lack of evidence for a link of between-subject variance in CMF and visual sensitivity for the VMA in Himmelberg, 2022 Nat Comms as opposed to the current finding. Are children vs adults extreme groups, affording better power? (cf de Haas, 2018 Front Hum Neurosci)

-> line 495: please clarify whether the studies you cite support the notion that polar angle asymmetries in visual performance *and cortical surface areas* are no longer present at intercardinal location, or whether this claim is specific to behaviour.

-> lines 642ff: please show example HRF fits for either group. What (canonical) HRF was used for the first pass of the fit? Did the eventual fits for children HRFs deviate more strongly from this assumed HRF for the first pass? Also, please provide descriptive stats on the proportion of vertices excluded ($R^2 < 10\%$) for either group.

Reviewer #3 (Remarks to the Author):

The manuscript describes a reanalysis of two existing data sets focusing on the HVA and VMA in both children and adults. The quantification of these asymmetries in children represents a novel demonstration despite the use of secondary data. The manuscript is well written and the authors have an excellent track records of high-quality research employing pRF modelling analyses.

I have only a few comments that the authors might wish to consider to strengthen the manuscript.

1. My main concern is the emphasis placed on similar median (and mean) values of explained variance. I would caution against overinterpreting the quality of the retinotopic maps on such a basis. As the authors know are surely aware, there can be non-unique solutions that the pRF model can return, which will have equivalent R2 values but different pRF estimates. It would be far more compelling (IMO) if the reliability in pRF estimates (Pa, Ecc, Size) were compared within subjects and then across groups. Presumably, there are sufficient runs in these original data sets in order to perform something like a split-half analysis?
2. Figure 6 puzzles me. Why show only six left and six right hemispheres? Would it not be more informative to show] both left and right hemispheres for all 12 children? If Figure size is an issue, these could be compared to a representative adult (as the adults show a typical pattern).
3. Paragraph starting line 379: The authors mention that the fMRI data do not 'perfectly' match the available behavioural data, but this is a cursory comment at best. However, this is a key point and should be discussed with appropriate depth and breadth.
4. Similarly, the section beginning on line 393 should include more discussion of why or how these identified changes might relate to the lack of VMA in children. At present it comes across as a list of interesting but somewhat unrelated developmental changes.

Comparing retinotopic maps of children and adults reveals a late-stage change in how V1 samples the visual field

Response to Reviewers

We would like to thank the reviewers for their positive assessment and constructive feedback on our manuscript. We have taken all their comments into consideration and our responses are below in blue text. Changes to the revised manuscript have been tracked in blue text and are noted below in italics with line numbers for reference. Addressing the reviewers' comments has strengthened our manuscript.

Best,

Marc Himmelberg, Postdoctoral Research Associate, NYU

Ekin Tünçok, Graduate Student, NYU

Jesse Gomez, Assistant Professor of Neuroscience, Princeton University

Kalanit Grill-Spector, Professor of Psychology, Stanford University

Marisa Carrasco, Julius Silver Professor of Psychology and Neural Science, NYU

Jonathan Winawer, Associate Professor of Psychology and Neural Science, NYU

Reviewer 1.

The ms reports very interesting data on the development of retinotopic asymmetry of primary visual cortex. In adult, in addition to horizontal and vertical meridian asymmetry, already well documented in the literature, are large, while the upper -lower vertical meridian asymmetry is small and difficult to document. These authors show that the vertical meridian asymmetry is not present in children 7-12 years old. Given the small effect the work is quite heroic. The presented data are convincing and clear: the polar maps are different between adults and children. However, I think that the result may be explained in different ways, and I hope that the authors could provide additional analysis that could dismiss these alternative explanations. I would have no hesitation in recommending publication given the implication of the results, if the author could provide this additional evidence.

1. The strongest evidence of the data are from the polar-map. This method relies on the strength of the BOLD responses. So if the lower field excitation produces a stronger BOLD inevitably the area associated with the stimulation will be larger (classical problem of circular statistic). The authors should demonstrate that the BOLD response of the lower and upper field have the same amplitude and the same S/N in children; and, crucially, also in adult (a clear problem that was not raised in the previous published paper).

We appreciate that the estimated size of a region of cortex responsive to a particular stimulus can depend on SNR and on choice of criterion. Both factors affect which (and how many) vertices to classify as being responsive to a stimulus. This is a confound in several fMRI papers which measure surface area of a region of interest by counting vertices that exceed threshold activation. However, we did not use this method in our study or in our previous work (Himmelberg, Winawer & Carrasco,

2022; Himmelberg, Kurzawski et al. 2021; Benson et al. 2021). All our analyses are based on pRF maps obtained from drifting bars, which are relatively insensitive to variation in BOLD amplitude, as noted by the reviewer in comment 2 below. We realize that some of our figures showing wedge-ROIs might have been ambiguous and could have been interpreted as if we had shown participants wedge stimuli in the scanner and measured the areas of cortex responsive to such stimuli. Such an experiment and analysis would indeed be subject to confounding factors such as variation in BOLD amplitude. However, we did not conduct such experiments or analyses. The fMRI experiments had drifting pRF bars and all analysis used pRF estimates.

We now include a new supplemental analysis to assess whether asymmetries in surface area are systematically related to asymmetries in variance explained (**Supplementary FIG. S13**; see also *Reviewer 2 Q2*). As expected, the two measures do not correlate, supporting the view that pRF estimates used to define our wedge-ROIs are not systematically biased by SNR.

Actions: Further clarification that our analyses are based on pRF centers (**Lines 217, 220-221, 769-770, FIG. 7 and 9 caption**). We have added a supplemental analysis showing that surface area asymmetries are not related to SNR (**Lines 344-349, Supplementary FIG. S13**).

2. The maps obtained from pRF with drift bars are less subject to the BOLD artefact. Unfortunately the authors did not attempt to generate an average pRF retinotopic map across participants to illustrate the map deformation.

Despite the appeal of an average retinotopic map as a summary figure, we chose to omit it because it can be misleading. Computing the average map requires co-registering the pRF maps to a single template. This registration introduces large distortions in surface area. Moreover, the standard templates are based on adult brains and may not be appropriate for children.

Action: Instead, we now present examples of all children's polar angle maps in native space in **FIG. 6** and **Supplementary FIG. S8** to illustrate the consistency of our data across all participants.

However, they evaluated the asymmetry by counting on individual participants the vertices associated with a particular orientation. The obtained results of fig S4 are not completely consistent with the results of the polar map, reinforcing the idea that BOLD amplitude may have played a role in estimating the asymmetry. Is the asymmetry difference significant in adults?

The asymmetry estimated from counting vertices (originally **Supplementary FIG S4**, which has now been removed from the resubmission) differed in many ways from the asymmetry estimated from surface area. One major factor was that not all vertices have the same surface area, and there are systematic biases such that vertices along the horizontal meridian tend to be smaller than those along the vertical meridian due to the geometry of the folded cortex. In turn we have removed this analysis as it is not a fair comparison and replaced it with a new one (**Supplementary FIG S6**). This analysis measures the surface area of vertices based on pRF centers, rather than continuous wedge-ROIs (This was also requested by *Reviewer 2*). This method is computationally simpler than the wedge-

ROI approach, but it is less resistant to noise. Both methods lead to the same pattern of results; a large HVA in children and adults, and a larger VMA in adults than children.

Action: New Supplementary FIG.S6.

The pRF model gives also pRF size, which may be different between upper and lower visual fields. These are important data that need to be presented.

We are also interested in pRF size as a function of age (children vs adults), and as a function of polar angle and eccentricity within individuals. pRF size as a function of eccentricity in children has been reported from the same dataset (see Gomez et al. 2018, *Nat Comms*). However, we now discuss several reasons why we chose to focus on surface area, rather than pRF size or other cortical measures (**Lines 366-385**).

Nonetheless, we have taken the reviewer's suggestion and computed pRF size as a function of eccentricity within polar angle wedges for children and adults and reported these values in **Supplemental Fig. S5**.

3. The stronger evidence in support of the asymmetry may derive from anatomical measures, not subject to BOLD response amplitude. In many visual pathologies also in children V1 thickness can change dramatically compared with typical participants. Please report the average thickness for the upper and lower vertical meridians. In this case the BOLD defined Roi can be used, given that it is an independent measure.

We agree that anatomical measures are important. Surface area provides such an anatomical measure –the surface area of a single vertex is derived from anatomical T1 data– and is not confounded by BOLD amplitude. Nor are the pRF centers used to define our wedge-ROIs (see *Reviewer 1 Q1*).

We note that estimates of cortical thickness can vary for different reasons (i.e., myelination differences masquerading as thickness differences (Natu et al. 2019; *PNAS*)). Thickness differences can also be ambiguous to interpret. As noted by the reviewer, cortex often thins in pathology and is associated with poorer functioning. In healthy brains, cortical thinning in V1 (or at least estimated thinning) is often associated with *better visual* performance (Song, Schwarzkopf, Kanai, & Rees 2015; *Neuron*). Therefore, we relied on surface area rather than cortical thickness or volume as our primary anatomical measure. Nonetheless, the estimated thickness values might be of interest, and we now report this, alongside measurements of curvature (since thickness and curvature are correlated throughout the brain).

Actions: Added cortical thickness and curvature estimates to **Supplementary FIGS. S3 and S4** which are noted on **Line 247-250**. See **Lines 366-385** for discussion of why surface area is our primary measure.

4. 24 participants is a large sample to attempt correlation analysis. The asymmetry of the horizontal and vertical meridian and of the upper and lower vertical meridian varies greatly between participants and this allows a correlative analysis with psychophysical thresholds. I suppose that many participants are the same of those of the recent *Current Biology* paper. If not these measures are very easy to collect in children, even with web based procedures. A correlation with psychophysics would greatly strengthen the results.

We had planned to collect both fMRI and psychophysical measurements from the same children (like our project with adults; Himmelberg, Winawer & Carrasco, 2022, *Nature Comms*), but Covid prevented us from doing so. The data in the current study are taken from prior work (Gomez et al. 2018 *Nat Comms*) and were collected at Stanford University in 2015. Thus, we do not have complementary psychophysical thresholds to correlate, and the children are 8 years older now. We plan to collect both fMRI and psychophysics with a new cohort of children. This will be a major undertaking and is beyond the scope of this project.

5. Eye movements are never mentioned or discussed in the manuscript. I understand that not all scanners have the capability to control fixation but at least the issue should be raised. The larger and noisier foveal confluence could be due to increased frequency of saccades in children?

The data used here were obtained from a prior study (Gomez et al. 2018, *Nature Comms*) at Stanford. Eye tracking was aborted in children if it took more than 5 minutes to set up, to avoid overly long scan sessions. But even in those cases, eye position was monitored by the IR camera. Any scans in which children broke fixation more than twice were restarted. We now note this in our methods (**Lines 623-629**). Eye tracking was only conducted in a limited number of adults (6 of 26), limiting statistical power. Nonetheless, the prior analysis of eye tracking data reported no significant difference in fixation behavior between adults and children (see Gomez et al. 2018 *Nature Comms*; first paragraph of Results and associated data in their Supplementary FIG 1).

We suspect that noise in the foveal confluence, in both adults and children, is partly due to the use of the pRF bar stimulus alone rather than a combination of bars, wedges, and rings, as we reported recently (Himmelberg, Kurzawski et al. 2020, *NeuroImage*). For this reason, we did not include data with eccentricity below 1° in all analyses of polar angle asymmetries.

Action: We amended the methods to describe eye tracking (**Lines 623-629**).

6. Could eye movement direction induce the vertical meridian asymmetry, endorsing a stronger influence of the motor system in the lower visual field representation in V1?

We do not think this is a plausible explanation First, in the studies showing a behavioral VMA for adults (but not children) eye fixation has been monitored and did not explain the polar angle asymmetries (e.g., Himmelberg, Winawer & Carrasco, *JoV* 2020; Barbot, Xue & Carrasco, *JoV* 2021; Carrasco et al., *Curr Bio* 2022). Second, the adult cortical VMA has been documented in 5 datasets (including this one). It is a reliable and consistent effect and unlikely to be an artifact of eye movements. Eye movements are unlikely to explain the lack of VMA in children because only scans in which children maintained fixation throughout the full run were used in the current study, as

indicated in the revision (see above). Third, prior work using simulated eye movements shows that even very large eye movements (4x larger than usual) have little effect on the estimated pRF centers (though they do increase pRF size) (see Levin et al., 2010; *Neuron*, FIG 6). In our analyses, only the pRF centers are used to define the wedge-ROIs that are then used to compute the cortical polar angle asymmetries, so it is unlikely that eye movements would abolish the child VMA as they would not cause any systematic change to the wedge-ROI definitions.

7. The fact that overall surface area of V1 and V2 are equal in children and adult, a greater representation of the lower visual field should move away from the fundus of V1 the representation of the horizontal meridian. Can please mark the fundus of calcarine sulcus on the retinotopic map?

The surface area of all of V1 is, on average, 115 mm² larger in adults than children. This is a small (non-statistically significant) difference compared to the total size of V1 (> 2,000 mm²) and compared to the within-group variability (about 2-fold difference from largest to smallest). Nonetheless, this mean difference is not small when compared to the difference in the surface area of the lower vertical meridian between children and adults. Even for the largest wedge-ROIs (45°), the lower vertical meridian is only about 100 mm² larger in adults than in children (~320 vs 220 mm²). This means that even though the lower vertical meridian representation is larger in adults than in children, it does not imply a dramatic change in the geometry of V1.

For illustration, we have provided an example child and adult with the fundus marked. The fundus is denoted as the black line through V1. In both cases, the fundus is close to the horizontal meridian.

8. 5 to 12 is a rather large age range. Do the older children show some anisotropy?

We are interested in the developmental trajectory of the VMA, but with a large age range of 5-12 there is limited power to see a correlation with age given the small number of participants at each age.

Action: Nonetheless, we have conducted an additional analysis correlating the magnitude of the cortical HVA and VMA with age for children (**Lines 308-310, Supplementary FIG S9**) and found no statistically reliable correlation.

9. Line 72. Cortical magnification here needs to be specified. Overall, the main point of the paper is to demonstrate a different cortical magnification for upper and lower visual field!

Done. See **Line 62 and 179-180**.

10. Fig 1 Please give the R^2 of the fit of the map. In adults the asymmetry is NOT very evident!!!

Done. See **FIG. 1** caption.

11. Fig3 A and B have the same y label, generating a lot of confusion.

The y-axis label of **FIG. 3A** is surface area in mm^2 , whereas **FIG. 3B** is the surface area in m^2 .

Action: We have amended the figure caption to explicitly state that the y-axis of **FIG 3A** is millimeters and **FIG. 3B** is meters.

12. 175 “within an eccentricity ring” – please rephrase the unclear sentence

We now explain the eccentricity ring (i.e., an annulus defined in visual space) in text (**Lines 182-186**) and we now refer to the methods where the eccentricity ring used to calculate cortical magnification with eccentricity is described in detail.

13. 544 please report the stimulus drift velocity.

Done. See **Line 617**.

Reviewer 2.

This paper builds on three recent sets of findings: 1) Children and adults show a visual sensitivity advantage for stimuli on the horizontal compared to the vertical meridian (the HVA). Adults also show an advantage for the lower over the upper vertical meridian, (the VMA), but -crucially – this is not seen for children (Carrasco et al., 2022 *Curr Biol*). 2) Individual differences in the adult HVA, but not the VMA correlate with corresponding anisotropies in cortical magnification (Himmelberg et al., 2022 *Nat Comms*). 3) The functional layout of early visual cortex in children appears adult like at an early age (e.g., Dekker et al., 2019 *Dev Cogn Neurosci*). However, previous studies on the development of retinotopic maps in children have not considered polar angle anisotropies in cortical magnification (CMF). The current paper sets out to fill this gap using fMRI and population receptive field mapping. The authors indeed find clear HVA and VMA for V1 cortical magnification in adult V1, but only an HVA for children, matching recent findings in psychophysics. This finding links between-subject variance in cortical magnification to that in visual sensitivity, which so far has not been possible for the VMA (Himmelberg et al., 2022). Additionally - and more importantly - it provides potential evidence for unexpectedly late large-scale changes in the functional organization of early visual cortex. The latter has wide ranging implications for visual neuroscience and general mechanisms of cortical organization, rendering the finding appealing for a wide audience. The hypotheses, experimental logic, writing and evidence are exceptionally clear. The methods appear solid and the authors took particular care to control for potential confounds by motion artefacts in children. However, I would like to suggest a few additional controls and know the authors' opinion on possible alternative interpretations of their findings.

1. AGE VS COHORT EFFECTS. Given you compare cross-sectional samples, could the findings be due to cohort effects rather than age? Specifically, children aged 5-12 today probably have had more (small) screen time than current 20-somethings had at the same age. Younger cohorts potentially also spent less time outdoors (as suggested by the myopia epidemic). Please discuss.

It is possible that the children had more screen time than adults had when they were children, but we see no linking hypothesis regarding how this could explain the results. Polar angle asymmetries have been identified in behavioral data going back 20 years, even though it is likely that the current adults had more screen time when they were children than those reported in older papers (e.g., Carrasco et al., 2001; Cameron et al. 2002; Talgar & Carrasco, 2002 vs. Barbot et al., 2021; Himmelberg et al. 2020, 2022).

Action: We now note that the asymmetries are unlikely to be cohort effects and the benefits of a longitudinal design (Lines 466-474)

2. POTENTIAL GROUP DIFFERENCES IN SPATIAL VARIANCE OF SNR. I think it is necessary to control for group differences in location-varying SNR effects, such as those induced by draining veins (Winawer et al., 2010 JoV). Relative to adults, is there a dip in the R2 map of children around the lower vertical meridian?

The draining vein issue in Winawer et al., (2010) obscured part of the hV4 map. Importantly, it was not known (and is still not agreed upon) what is on the other side of the hV4 map. So, obscuring that part of the map could actually make estimates of map size smaller. Here, we know what is on either side of the V1 map: dorsal and ventral V2, and we could identify the polar angle reversal in each participant. This means that there is no missing part of the maps that are potentially reducing our estimates of surface area.

Overall, we developed a method of computing surface area that is robust to SNR (see **Lines 87-90, 787-770, 844-846**). We use stimulus referred parameters (polar angle, eccentricity) to define wedge-ROI width, and we use the measures in a way that is largely resistant to systematic biases. The method entailed estimating enclosed, contiguous regions of cortex for each visual field wedge-ROI, rather than the speckly representation one would get from measuring each individual vertex whose pRF was within some polar angle boundary. The latter would be subject to SNR-dependent biases, as parts of the visual field that had lower SNR might have fewer vertices included. Our main analysis did not use this vertex-counting method (although we now included a similar analysis in **Supplementary FIG. S6**). Even if vertices had a relatively low SNR (R^2 lower than 10%), they were *included in the wedge-ROI* (see **Lines 844-846**) To be more conservative, in our summary calculations of the asymmetries (**FIG. 7**), we now analyze wedge-ROIs defined by 25° rather than 15° .

Nonetheless, we have calculated the median R^2 of the vertices within the upper vertical and lower vertical meridian wedge-ROI (25°). The values are high (50-60%) compared to typical R^2 thresholds (10%), so it seems that we are not losing data. Moreover, the LVM has a numerically higher R^2 than the UVM in children, implying that there is no special loss of lower meridian representation in children causing it to be smaller than it is in adults.

We note that Reviewer 1 also raised questions about variance explained and SNR. See their comment and our response (*Reviewer 1, comment 2*).

Further, we have included a new supplemental analysis (**Supplementary FIG. S13**) to assess whether asymmetries in surface area are systematically related to asymmetries in variance explained.

Action: We changed the wedge-ROI width for computing polar angle asymmetries, we are now explicit that our analyses do not depend systematically on SNR or data quality at the beginning of our results (**Lines 87-90**), and we include **Supplementary FIG. S13**.

3. POTENTIAL IMPACT OF PRIORS AND TEMPLATES ON RESULTS (lines 706ff). The potential impact of the Bayesian eccentricity template and polar angle cleaning algorithm seem important, esp. given templates probably are derived from adult data(?). To what degree do the results hinge on these processing steps? Does the general pattern of results replicate when only using the empirical data, e.g. by calculating the sum of cortical area for all vertices with pRFs falling into a given visual field ROI, regardless of map discontinuities (this may be improved by simple surface smoothing)? If this is too noisy, is there any other way of ensuring the pattern of results isn't imposed onto the data by algorithms imposing prior assumptions?

The Bayesian eccentricity measurement combines unique participant data with an adult eccentricity template. However, eccentricity measurements between adults and children are similar, thus this template is appropriate for use with children. The main goal of the eccentricity template is to ensure that the data come from a contiguous eccentric portion of V1. The cleaning optimization was implemented on the pRF polar angle fits in V1. This minimization seeks to adjust the pRF centers of vertices as little as possible while simultaneously enforcing a smooth cortical magnification map and correcting the field sign values across V1; this method does not rely on any template and uses unique participant data. It is essentially smoothing the data and does so similarly for adults and children.

Action: We now include a supplementary analysis in which we measured the surface area of V1 dedicated to portions of the visual field without enforcing contiguous wedge-ROIs using raw polar angle and eccentricity values, thus no templates (See **Lines 2552-259, Supplementary FIG. S6**). Our results still hold between 15-35° of angle –the measurements closest to the meridian.

4. CHANGES IN STRUCTURE VS CHANGES IN pRF MAPPING ONTO STRUCTURE. The distinction discussed in lines 433ff seems relevant. Isn't it possible to juxtapose the two hypotheses with the data you have at hand? If the surface area of the upper bank of the calcarine expands disproportionately in adults (the 'structure' hypothesis), this should become apparent by comparing the surface area of the upper and lower bank between groups, only using the meridian lines to define them (comparing whole quarter fields rather than polar wedges). However, if the structure stays the same and only pRFs migrate within the V1 map (the 'mapping' hypothesis), such an analysis should not yield any group differences. But maybe I'm missing something?

We proposed two explanations for the increase in surface area for the lower vertical meridian in adults. One is a change in receptive field location and the other is tissue growth around the lower vertical meridian neural population. Both predict estimates of greater surface area near the lower vertical meridian. The comparison of entire quarterfields is already included in **FIG. 5B** and **5D** as the $\pm 45^\circ$ point on the x-axis. Note that the $\pm 45^\circ$ wedge-ROI covers a total of 90° of visual space. For the

$\pm 45^\circ$ wedge-ROI, there is a large difference between the upper and lower field in adults, but not in children. Whereas it might be possible to formulate quantitative, geometric hypotheses about the changes in maps between adults and children, that is outside the scope of this paper. We think it is an interesting topic but one that could be studied as a follow-up now that we are reporting the primary results revealing a difference. See also response to *Reviewer 1, Question 7*.

Action: We now address this more explicitly and note that longitudinal data would help discriminate between the two possibilities (**Lines 522-525**).

5. Figures S7 and S8 suggest that the adult VMA in V1 is entirely driven by the close parafoveal range below 2 or 3 degrees of eccentricity. I think this finding deserves more attention and should be discussed, maybe in the context of retinal developments in line 400ff. Is this a real effect or more likely an artefact of data attrition for higher eccentricity bands? What's the evidence for the perceptual VMA at different eccentricity bands?

There is an imperfect match between cortical measures and behavioral measures; the perceptual HVA and VMA become stronger with eccentricity for adults (Carrasco et al. 2001, *Spat Vis*), but cortical measures in **Supplementary FIGS. S10** and **S11** do not. We do not know the explanation; it could be due to measurement noise at higher eccentricities (less vertices to make surface area estimates), due to fMRI and behavioral measurements being collected from separate groups, or to a true lack of correlation between brain and behavior with respect to these variables.

Action: We now expand our discussion of this point (**Lines 446-451**).

6. Given the broad age range among children, is there a trend for the VMA emerging in older children? It would be nice to see a scatter plot of VMA vs age.

Action: We have included an additional analysis correlating the magnitude of the cortical HVA and VMA with age for children (**Lines 309-310, Supplementary FIG. S9**). We find no statistically reliable correlation between age and VMA.

7. Please show some example time-series and fits from adult and children's vertices (e.g. in Figure 1)

Action: Examples of the pRF fits to the BOLD signal from a V1 vertex for a sample adult and child are available in new **FIG 8**.

8. Lines 130ff: the high degree of similarity between children and adults is very apparent in Figure 3. However, I don't think a 'p>0.5' is informative or appropriate in this context. If using tools from inference statistics, I suggest to report the confidence interval for a mean difference between groups instead to show which effect sizes can be excluded as unlikely (mean differences probably best expressed in % of adult total surface area).

Action: We have amended these results to show bootstrapped percent difference in map size between adults and children. We report the mean percent change and confidence intervals for these comparisons (**Lines 137-142**). Note that **FIG 3**. still contains raw data to show variability in map size.

9. Lines 162ff I appreciate the efforts to control for potential motion confounds. However, I don't quite get the point of this analysis. Is there any reasonable expectation that surface area estimates would correlate with motion? If so, please briefly explain the reasoning. Similarly for lines 313ff. Also, it would be good to give the reader the descriptive stats from Figure S1 here already (median number of frames with excessive motion 1 for adults, 10 for children). Regarding the latter, I think the methods section fails to state the number of volumes (only four runs mentioned, unless I missed it). Please add for context.

We agree with the reviewer's intuition that there is no obvious link between head motion and surface area. Given the surprising result that children's VMA differs from adults', we felt it prudent to check whether possible differences in motion would correlate with any of our measures (especially because motion is a measurement that differs between the two groups in our dataset). We have now justified, condensed, and simplified these checks to show that 1) children move more than adults, 2) however motion does not impact surface area measurements of V1-V3, and 3) there is no systematic relationship between observer SNR and the magnitude of their VMA.

Action: Previously we had motion correlation analyses in 3 different parts of the Results. We have now placed all the safety checks in a single supplemental section (**Supplementary FIGS. S12 and 13 and related text**). This is noted on **Lines 332-349**.

Lines 335-337: We now provide descriptive statistics (median number of frames with excessive motion) for **FIG. S12A**.

Line 644: Added. 96 volumes per image.

10. What central tendency do the lines in Figure 4 show? Means? Medians? Please add

Action: Amended. **FIG. 4** caption now identifies the lines as means across participants.

11. It may be worth explicitly discussing possible reasons for the lack of evidence for a link of between-subject variance in CMF and visual sensitivity for the VMA in Himmelberg, 2022 Nat Comms as opposed to the current finding. Are children vs adults extreme groups, affording better power? (cf de Haas, 2018 Front Hum Neurosci)

We agree that a likely explanation is that the comparison with children greatly increases the dynamic range. We have a correlation at the group level (adults show cortical and behavioral VMA, whereas children show neither) but not at the individual level within adults (no correlation between cortical and behavioral VMA for adults, as reported in Himmelberg 2022, *Nat Comms*).

Action: We now make this point explicit on **Lines 457-464**.

12. line 495: please clarify whether the studies you cite support the notion that polar angle asymmetries in visual performance *and cortical surface areas* are no longer present at intercardinal location, or whether this claim is specific to behaviour.

Lines 555-557: Clarified. The studies show that the polar angle asymmetries for *both* visual performance and cortical surface are largest at the meridians and gradually decrease to the point that they are often no longer present at intercardinal locations.

13. lines 642ff: please show example HRF fits for either group. What (canonical) HRF was used for the first pass of the fit? Did the eventual fits for children HRFs deviate more strongly from this assumed HRF for the first pass? Also, please provide descriptive stats on the proportion of vertices excluded ($R^2 < 10\%$) for either group.

New **FIG. 8** (related text on **Line 729-731**) contains plots of the mean child and adult final HRF fits, and the two-gamma starting HRF. Adults and children have a similar peak, however, undershoot is slightly larger for adults.

Action: We specify that two gamma HRF was used for the first pass of the fit (**Line 711**).

Regarding the proportion of vertices with $R^2 < 0.10$. First, we note that very few vertices within V1 have pRF variance explained less than 10%. This is evident in several figures showing retinotopic maps, including a new figure showing the maps of all 25 children: these maps are thresholded at 10% and one can see that there are almost no holes in any of the maps, meaning there is very little data subject to this threshold.

Moreover, whereas the 10% threshold was used for several analyses, it was *not* used for the most important analysis in the manuscript, the measurements of surface area *within* wedge-ROIs. These analyses did not exclude any vertices due to low R^2 . The analyses which did depend on the threshold are (1) selecting voxels for optimizing the hRF fit; (2) the alternative method for computing surface area by vertex (i.e., without enforcing contiguity); (3) selecting vertices to use to find the iso-angle boundaries of the wedge-ROIs (but not which vertices were included *within* that wedge-ROI); (4)

hand-drawing map boundaries. Thus, the primary analyses of the paper (surface area within wedge-ROIs) did not exclude vertices.

Action: We now state clearly which analyses used the 10% threshold **FIG 1 caption, Lines 747, 817, 844-846.**

Nonetheless, because some analyses did exclude vertices with pRF $R^2 > 10\%$, we provide here a table summarizing the number of such vertices in V1, separately for different wedge ROIs. The numbers are small, ranging from a fraction of a percent to less than 3%.

	$\pm 15^\circ$	$\pm 25^\circ$	$\pm 35^\circ$	$\pm 45^\circ$	$\pm 55^\circ$
Adults	0.25%	0.17%	0.14%	0.06%	0.09%
Children	2.6%	1.2%	1.32%	0.78%	1.14%

Table 1. Mean proportion of vertices within each wedge-ROI whose pRF variance explained was below 10%.

Reviewer 3.

The manuscript describes a reanalysis of two existing data sets focusing on the HVA and VMA in both children and adults. The quantification of these asymmetries in children represents a novel demonstration despite the use of secondary data. The manuscript is well written and the authors have an excellent track record of high-quality research employing pRF modeling analyses. I have only a few comments that the authors might wish to consider to strengthen the manuscript.

1. My main concern is the emphasis placed on similar median (and mean) values of explained variance. I would caution against overinterpreting the quality of the retinotopic maps on such a basis. As the authors are surely aware, there can be non-unique solutions that the pRF model can return, which will have equivalent R2 values but different pRF estimates. It would be far more compelling (IMO) if the reliability in pRF estimates (Pa, Ecc, Size) were compared within subjects and then across groups.

The goal of our data quality analyses was to show that both adults and children have high quality retinotopic maps –it was not our intention to show that the quality of the maps are identical. The primary measures we report –surface area of whole maps or parts of maps– are not systematically affected by data quality, as long as there is sufficient quality to identify the maps.

Prior work (Lerma-Usabiaga, Benson, Winawer & Wandell, 2020, *Plos Comp Bio*) shows that whereas there can be high uncertainty with regards to pRF sizes (i.e. different pRF sizes may have similar variance explained for the same voxel time series), for our stimulus sequence and analysis method, there is good reliability for estimating polar angle and eccentricity (see also Himmelberg, Kurzawski et al. 2021 *NeuroImage*). None of our key surface area analyses depend upon pRF size, the least certain parameter in the pRF fit.

Action: We are now explicit that we are assessing the overall quality of the maps, rather than comparing quality between groups (**Lines 85-92**). We now show individual retinotopic maps in both hemispheres of all 25 children (**FIG. 6 and Supplementary FIG. S8**) and all left hemispheres adults (**FIG. S6**); the maps show specific structure attributes (e.g., horizontal meridian in the calcarine sulcus; smooth gradations of polar angle, etc) that indicate good quality data. We now make it explicit in the manuscript methods that we used a 5-step fitting process to fit the pRF model, that minimizes the chances of fitting noise, rather than the underlying fMRI signal (**Lines 721-723**).

Presumably, there are sufficient runs in these original data sets in order to perform something like a split-half analysis?

With respect to the question about split halves: It is possible to run a split half analysis on our data; however, each split would only comprise 2 short scans (96 TRs) thereby compromising SNR so we do not think this would give us an accurate metric of the data quality.

Action: We now make explicit that we show a correlation between the left and right surface area of the V1-V3 ROIs, a test that shows the data are of sufficient quality to estimate map size in both kids and adults (**Lines 117-118**).

2. Figure 6 puzzles me. Why show only six left and six right hemispheres? Would it not be more informative to show] both left and right hemispheres for all 12 children? If Figure size is an issue, these could be compared to a representative adult (as the adults show a typical pattern).

Action: FIG. 6 now includes one example adult map and left hemisphere maps for all 25 children. Right hemisphere maps for all 25 children are available in **Supplementary FIG. S8** and all 24 left hemisphere adult maps are available in **Supplementary FIG S7**.

3. Paragraph starting line 379: The authors mention that the fMRI data do not 'perfectly' match the available behavioural data, but this is a cursory comment at best. However, this is a key point and should be discussed with appropriate depth and breadth.

Action: Lines 441-455: We have now clarified and expanded this point.

4. Similarly, the section beginning on line 393 should include more discussion of why or how these identified changes might relate to the lack of VMA in children. At present it comes across as a list of interesting but somewhat unrelated developmental changes.

Agreed. We already have a section exploring possible biological underpinnings of the change in VMA between children and adults (**Lines 476-525**), and many of the studies in this section were already referred to and discussed in other places in the manuscript. Therefore, we think that the section the reviewer refers to is no longer necessary.

Action: We deleted this section.

REVIEWERS' COMMENTS

Reviewer #1 (Remarks to the Author):

The authors performed a thoroughly revision, adding important new analyses that dismissed all my concerns. I advise acceptance in the present form.

Reviewer #2 (Remarks to the Author):

I thank the authors for thoroughly addressing all points I raised and congratulate them to this excellent piece of work.

Reviewer #3 (Remarks to the Author):

The authors have addressed my concerns and believe the edits in response to all reviewers has strengthened the manuscript. I congratulate the authors on yet another impressive piece of research.

Comparing retinotopic maps of children and adults reveals a late-stage change in how V1 samples the visual field

Response to Reviewers

We would like to thank the reviewers for their positive assessment on our revised manuscript. The reviewers had no further comments on our manuscript. We have included our responses to the editorial requests in the Author checklist document, and all requested changes have been made in blue text in the manuscript.

Best,

Marc Himmelberg, Postdoctoral Research Associate, NYU

Ekin Tünçok, Graduate Student, NYU

Jesse Gomez, Assistant Professor of Neuroscience, Princeton University

Kalanit Grill-Spector, Professor of Psychology, Stanford University

Marisa Carrasco, Julius Silver Professor of Psychology and Neural Science, NYU

Jonathan Winawer, Associate Professor of Psychology and Neural Science, NYU